# SPEED: Scalable, Precise, and Efficient Concept Erasure for Diffusion Models

**Ouxiang Li[1]\*, Yuan Wang[1]\*, Xinting Hu[1]†, Houcheng Jiang[1], Jack Tao[1],**
**Yanbin Hao[2], James Ma[1], Fuli Feng[1]†**

[1]University of Science and Technology of China, [2]Hefei University of Technology
`lioox@mail.ustc.edu.cn, joyhu1412@gmail.com, fulifeng93@gmail.com`

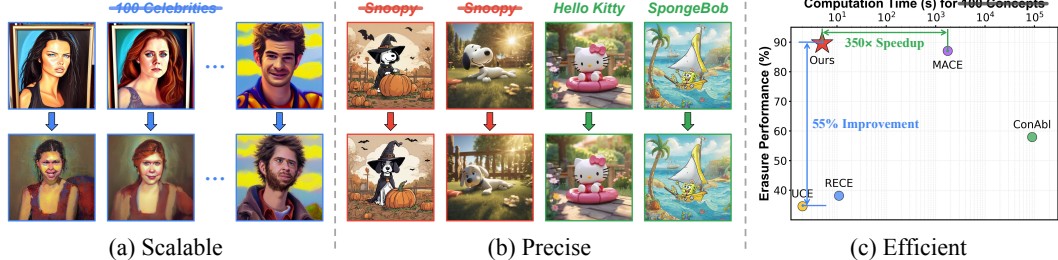

(a) Scalable      (b) Precise      (c) Efficient

Figure 1: **Three characteristics of our proposed concept erasure method for diffusion models, SPEED.** **(a) Scalable:** SPEED seamlessly scales from single-concept to large-scale multi-concept erasure (*e.g.*, 100 celebrities) without additional design. **(b) Precise:** SPEED precisely removes the target concept (*e.g.*, *Snoopy*) while preserving the semantics for non-target concepts (*e.g.*, *Hello Kitty* and *SpongeBob*). **(c) Efficient:** SPEED immediately erases 100 concepts within 5 seconds, achieving new state-of-the-art (SOTA) performance with a $350\times$ speedup over competitive methods.

## ABSTRACT

Erasing concepts from large-scale text-to-image (T2I) diffusion models has become increasingly crucial due to the growing concerns over copyright infringement, privacy violations, and offensive content. In scalable erasure applications, fine-tuning-based methods are time-consuming to precisely erase multiple target concepts, while real-time editing-based methods often degrade the generation quality of non-target concepts due to conflicting optimization objectives. To address this dilemma, we introduce SPEED, a *scalable, precise, and efficient* concept erasure approach that directly edits model parameters. SPEED searches for a null space, a model editing space where parameter updates do not affect non-target concepts, to achieve scalable and precise erasure. To facilitate accurate null space optimization, we incorporate three complementary strategies: Influence-based Prior Filtering (IPF) to selectively retain the most affected non-target concepts, Directed Prior Augmentation (DPA) to enrich the filtered retain set with semantically consistent variations, and Invariant Equality Constraints (IEC) to preserve key invariants during the T2I generation process. Extensive evaluations across multiple concept erasure tasks demonstrate that SPEED consistently outperforms existing methods in non-target preservation while achieving efficient and high-fidelity concept erasure, successfully erasing 100 concepts within only 5 seconds. Our code and models are available at: *https://github.com/Ouxiang-Li/SPEED*.

## 1 INTRODUCTION

Text-to-image (T2I) diffusion models (Ho et al., 2020; Song et al., 2020a;b; Nichol & Dhariwal, 2021; Rombach et al., 2022; Ho & Salimans, 2022) have facilitated significant breakthroughs in generating highly realistic and contextually consistent images simply from textual descriptions (Dhariwal & Nichol, 2021; Ramesh et al., 2021; Gal et al., 2022; Betker et al., 2023; Ruiz et al., 2023;

---

\*Equal Contributions.
†Corresponding authors.

Podell et al., 2023; Esser et al., 2024; Xu et al., 2025a;b). Alongside these advancements, concerns have also been raised regarding copyright violations (Cui et al., 2023; Shan et al., 2023; Yan et al., 2024), privacy concerns (Carlini et al., 2023; Yang et al., 2023), and offensive content (Schramowski et al., 2023; Yang et al., 2024b; Zhang et al., 2025). To mitigate ethical and legal risks in generation, it is necessary to prevent the model from generating certain concepts, a process termed ***concept erasure*** (Kumari et al., 2023; Gandikota et al., 2023; Zhang et al., 2024a). However, removing target concepts without carefully preserving the semantics of non-target concepts can introduce unintended artifacts, distortions, and degraded image quality (Gandikota et al., 2023; Orgad et al., 2023; Schramowski et al., 2023; Zhang et al., 2024a), compromising the model's usability. Therefore, beyond ensuring the effective removal of target concepts (*i.e.*, ***erasure efficacy***), concept erasure should also maintain the original semantics of non-target concepts (*i.e.*, ***prior preservation***).

In this context, recent methods strive to seek a balance between erasure efficacy and prior preservation, broadly categorized into two paradigms: training-based (Kumari et al., 2023; Lyu et al., 2024; Lu et al., 2024a) and editing-based (Gandikota et al., 2024; Gong et al., 2025). The training-based paradigm fine-tunes diffusion models to achieve concept erasure, incorporating an additional regularization into the training objective for prior preservation. In contrast, the editing-based paradigm avoids additional fine-tuning by directly modifying model parameters (*e.g.*, projection weights in cross-attention layers (Rombach et al., 2022)), with such modifications derived from a closed-form objective that jointly accounts for erasure and preservation. This efficiency also facilitates editing-based methods to extend to multi-concept erasure without additional designs seamlessly.

However, as the number of target concepts increases, current editing-based methods (Gandikota et al., 2024; Gong et al., 2025) struggle to balance between erasure efficacy and prior preservation. This can be attributed to the growing conflicts between erasure and preservation objectives, making such trade-offs increasingly difficult. Moreover, these methods rely on weighted least squares optimization, inherently imposing a ***non-zero lower bound*** on preservation error (see Appx. B.2). In multi-concept settings, this accumulation of preservation errors gradually distorts non-target knowledge, thereby degrading prior preservation. To address the above limitations, we propose a **S**calable, **P**recise, and **E**fficient Concept **E**rasure for **D**iffusion Models (SPEED) (see Fig. 1), an editing-based method with null-space constraints. Specifically, we search for the ***null space of prior knowledge***, a model editing space where parameter updates do not affect the feature representations of non-target concepts. By projecting the model parameter updates for concept erasure onto this null space, SPEED can minimize the preservation error to zero without compromising erasure efficacy, thereby enabling scalable and precise concept erasure without affecting non-target concepts.

The main contribution of SPEED lies in defining an effective null space from a set of predefined non-target concepts (*i.e.*, ***retain set***). We observe that the existing baseline with null-space constraints (Fang et al., 2024) confronts a fundamental dilemma during concept erasure: While a small retain set limits the coverage of prior knowledge, enlarging the retain set makes it increasingly difficult to identify an accurate null space. This dilemma arises because a large retain set drives the corresponding feature matrix toward full rank, necessitating the estimation of its null space to ensure sufficient degrees of freedom for concept erasure. However, such estimation inevitably perturbs the representations of the retain set concepts, leading to semantic degradation and compromised prior preservation, as illustrated in Fig. 2.

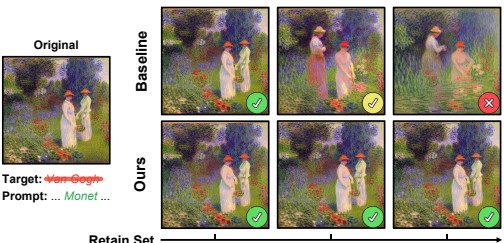

**Figure 2: Semantic degradation with increasing retain set concepts.** The baseline null-space constrained method (Fang et al., 2024) preserves non-target semantics well given a small retain set ✅. However, as the retain set concepts increase, the corresponding matrix approaches higher rank, making the null space estimation increasingly inaccurate (see Eq. 4) with inevitable approximation errors, thereby degrading prior preservation ✅❌.

In this light, we introduce ***Prior Knowledge Refinement,*** a suite of techniques designed to strategically and selectively refine the retain set so as to mitigate the semantic degradation in searching for the null space. Particularly, we propose *Influence-based Prior Filtering (IPF)*, which first quantifies the influence of concept erasure on each non-target concept and then prunes the retain set by removing minimally affected concepts, preventing the correlation matrix from approaching full rank and thus maintaining an accurate null space. Subsequently, to further enhance prior preservation over

the resulting retain set, we propose *Directed Prior Augmentation (DPA)*, which expands the retain set with directed, semantically consistent perturbations to improve retain set coverage. In addition, we incorporate *Invariant Equality Constraints (IEC)*, which impose equality constraints on invariant representations during generation (*e.g.*, [SOT] token), ensuring that they remain unchanged throughout concept erasure. We evaluate SPEED on three representative concept erasure tasks, *i.e.*, few-concept, multi-concept, and implicit concept erasure, where it consistently exhibits superior prior preservation across all erasure tasks. Our contributions can be summarized as follows:

- We propose SPEED, a scalable, precise, and efficient concept erasure method with null-space constrained model editing, capable of erasing 100 concepts in 5 seconds.
- We introduce Prior Knowledge Refinement to construct an accurate null space over the retain set for effective editing. Leveraging three complementary techniques, IPF, DPA, and IEC, our method balances semantic degradation and retain coverage, enabling precise and scalable concept erasure.
- Our extensive experiments show that SPEED consistently outperforms existing methods in prior preservation across various erasure tasks with minimal computational costs.

## 2 RELATED WORKS

**Concept erasure.** Current T2I diffusion models inevitably involve unauthorized and offensive generations due to the noisy training data from web (Schuhmann et al., 2021; 2022). Apart from applying additional filters or safety checkers (Rando et al., 2022; Betker et al., 2023; Rao, 2023), prevailing methods modify diffusion model parameters to erase specific target concepts, mainly categorized into two paradigms. The training-based paradigm fine-tunes model parameters with specific erasure objectives (Kumari et al., 2023; Gandikota et al., 2023; Zhang et al., 2024a;b; Zhao et al., 2024b;a; Huang et al., 2024; Kim et al., 2024; Wu et al., 2025a) and additional regularization. In contrast, the editing-based paradigm edits model parameters using a closed-form solution to facilitate efficiency in concept erasure (Orgad et al., 2023; Gandikota et al., 2024; Gong et al., 2025). These methods can erase numerous concepts within seconds, demonstrating superior efficiency in practice. Beyond parameter modification, non-parametric methods (*e.g.*, external modules and sampling interventions) have also been explored (Schramowski et al., 2023; Wang et al., 2024b; Yoon et al., 2024; Jain et al., 2024; Lee et al., 2025b;a), but they are fragile in open-source settings.

**Null-space constraints.** The null space of a matrix, a fundamental concept in linear algebra, refers to the set of all vectors that the matrix maps to the zero vector. The null-space constraints are first applied to continual learning by projecting gradients onto the null space of uncentered covariances from previous tasks (Wang et al., 2021). Subsequent studies (Lu et al., 2024b; Wang et al., 2024a; Yang et al., 2024a; Kong et al., 2022; Lin et al., 2022) further explore and extend the application of null space in continual learning. In model editing, AlphaEdit (Fang et al., 2024) restricts model weight updates onto the null space of preserved knowledge, effectively mitigating trade-offs between editing and preservation. Null-space constraints also apply to various tasks, *e.g.*, machine unlearning (Chen et al., 2024), MRI reconstruction (Feng et al., 2023), and image restoration (Wang et al., 2022), offering promise for editing-based concept erasure.

## 3 PROBLEM FORMULATION

In T2I diffusion models, each concept is encoded by a set of text tokens via CLIP (Radford et al., 2021), which are then aggregated into a single concept embedding $c \in \mathbb{R}^{d_0}$. For concept erasure, there are two sets of concepts: the erasure set $\mathbf{E}$ and the retain set $\mathbf{R}$. The erasure set consists of $N_E$ target concepts to be removed, denoted as $\mathbf{E} = \{c_1^{(i)}\}_{i=1}^{N_E}$. The retain set includes $N_R$ non-target concepts that should be preserved during editing, denoted as $\mathbf{R} = \{c_0^{(j)}\}_{j=1}^{N_R}$. To enable efficient erasure efficacy for $\mathbf{E}$ and prior preservation for $\mathbf{R}$, we first formulate a closed-form editing objective in Sec. 3.1, and enhance it with null-space constrained optimization in Sec. 3.2.

### 3.1 CONCEPT ERASURE IN CLOSED-FORM SOLUTION

To effectively erase each target concept $c_1^{(i)} \in \mathbf{E}$ (*e.g.*, ~~Snoopy~~), it is specified to be mapped onto an anchor concept $c_*^{(i)}$ that shares general semantics (*e.g.*, *Dog*), termed as an anchor set

$\mathbf{A} = \{c_*^{(i)}\}_{i=1}^{N_E}$. For editing-based methods (Orgad et al., 2023; Gandikota et al., 2024; Gong et al., 2025), concept embeddings from the erasure set $\mathbf{E}$, anchor set $\mathbf{A}$, and retain set $\mathbf{R}$ are first organized into three structured matrices: $\mathbf{C}_1, \mathbf{C}_* \in \mathbb{R}^{d_0 \times N_E}$ and $\mathbf{C}_0 \in \mathbb{R}^{d_0 \times N_R}$, representing the stacked embeddings of target, anchor, and non-target concepts, respectively. To derive a closed-form solution for concept erasure, existing methods typically optimize a perturbation $\mathbf{\Delta}$ to model parameters $\mathbf{W}$, balancing between erasure efficacy and prior preservation. For example, UCE (Gandikota et al., 2024) formulates concept erasure as a weighted least squares problem:

$$\mathbf{\Delta}_{\text{UCE}} = \arg\min_{\mathbf{\Delta}} \underbrace{\|(\mathbf{W} + \mathbf{\Delta})\mathbf{C}_1 - \mathbf{W}\mathbf{C}_*\|^2}_{e_1} + \underbrace{\|\mathbf{\Delta}\mathbf{C}_0\|^2}_{e_0}, \tag{1}$$

where the erasure error $e_1$ ensures that each target concept is mapped onto its corresponding anchor concept and the preservation error $e_0$ minimizes the impact on non-target concepts, and $\|\cdot\|^2$ denotes the sum of the squared elements in the matrix (*i.e.*, *Frobenius norm*). This formulation provides a closed-form solution $\mathbf{\Delta}_{\text{UCE}}$ (see Appx. B.1) for parameter updates, achieving computationally efficient optimization. However, as the number of target concepts increases, the accumulated preservation errors $e_0$, which admit a provable non-zero bound (see Appx. B.2), across multiple target concepts would amplify the impact on non-target knowledge and degrade prior preservation.

## 3.2 Apply Null-Space Constraints

To address the limitation of weighted optimization in prior preservation, SPEED incorporates null-space constraints (Wang et al., 2021; Fang et al., 2024) to achieve prior-preserved model editing by forcing $e_0 = 0$. The null space of $\mathbf{C}_0$ consists of all vectors $\boldsymbol{v}$ such that $\boldsymbol{v}\mathbf{C}_0 = \mathbf{0}$. Restricting the parameter update $\mathbf{\Delta}$ to this space ensures that such updates do not interfere with non-target concepts.

To project $\mathbf{\Delta}$ onto null space, we apply singular value decomposition (SVD) on $\mathbf{C}_0\mathbf{C}_0^\top \in \mathbb{R}^{d_0 \times d_0}$ [1] and have $\{\mathbf{U}, \mathbf{\Lambda}, \mathbf{U}^\top\} = \text{SVD}(\mathbf{C}_0\mathbf{C}_0^\top)$, where $\mathbf{U} \in \mathbb{R}^{d_0 \times d_0}$ contains the singular vectors of $\mathbf{C}_0\mathbf{C}_0^\top$, and $\mathbf{\Lambda}$ is a diagonal matrix of its singular values. The singular vectors in $\mathbf{U}$ w.r.t. zero singular values form an orthonormal basis for the null space of $\mathbf{C}_0$, which we denote as $\hat{\mathbf{U}}$. Using this basis, we construct the null-space projection matrix $\mathbf{P} = \hat{\mathbf{U}}\hat{\mathbf{U}}^\top$. This process is formulated as:

$$\{\mathbf{U}, \mathbf{\Lambda}, \mathbf{U}^\top\} = \text{SVD}(\mathbf{C}_0\mathbf{C}_0^\top), \quad \hat{\mathbf{U}} = \mathbf{U}[:, \mathbf{\Lambda}_{ii} = 0], \quad \mathbf{P} = \hat{\mathbf{U}}\hat{\mathbf{U}}^\top. \tag{2}$$

The final update applied to model parameters is $\mathbf{\Delta}\mathbf{P}$, which projects $\mathbf{\Delta}$ onto the null space of $\mathbf{C}_0$. This ensures that updates do not interfere with non-target concepts, satisfying $\|(\mathbf{\Delta}\mathbf{P})\mathbf{C}_0\|^2 = 0$. To solve for the updates, we minimize the following objective:

$$\mathbf{\Delta}_{\text{Null}} = \arg\min_{\mathbf{\Delta}} \underbrace{\|(\mathbf{W} + \mathbf{\Delta}\mathbf{P})\mathbf{C}_1 - \mathbf{W}\mathbf{C}_*\|^2}_{e_1} + \underbrace{\|(\mathbf{\Delta}\mathbf{P})\mathbf{C}_0\|^2}_{e_0 = 0} + \underbrace{\|\mathbf{\Delta}\mathbf{P}\|^2}_{\text{regularization}}, \tag{3}$$

where $\|\mathbf{\Delta}\mathbf{P}\|^2$ is a regularization term to ensure convergence. The preservation term $\|(\mathbf{\Delta}\mathbf{P})\mathbf{C}_0\|^2$ is omitted, as it is guaranteed to be zero by the null-space constraint ($e_0 = 0$). This objective allows the model parameters to be updated such that target concepts are effectively erased while representations of non-target concepts remain unaffected, thereby achieving prior-preserved concept erasure.

## 4 Prior Knowledge Refinement

However, as more diverse non-target concepts are included in the retain set, the rank of the correlation matrix $\mathbf{C}_0\mathbf{C}_0^\top$ gradually increases [2]. The null space, defined as the orthogonal complement of this span, correspondingly shrinks in dimension:

$$\dim(\text{Null}(\mathbf{C}_0\mathbf{C}_0^\top)) = d_0 - \text{rank}(\mathbf{C}_0\mathbf{C}_0^\top). \tag{4}$$

Here, the null space dimension characterizes the degrees of freedom available for editing without affecting the non-target concepts. However, as this dimension shrinks, to ensure sufficient degrees of

---

[1] $\mathbf{C}_0\mathbf{C}_0^\top$ and $\mathbf{C}_0$ share the same null space. We operate on $\mathbf{C}_0\mathbf{C}_0^\top \in \mathbb{R}^{d_0 \times d_0}$ since it has fixed row dimension while $\mathbf{C}_0 \in \mathbb{R}^{d_0 \times N_R}$ may have high dimensionality depending on concept number $N_R$.

[2] We assume that the concepts are not exactly linearly dependent in the representation space, which is generally satisfied in practice due to the semantic diversity and high dimensionality of the embedding space.

freedom for concept erasure, we are compelled to include additional singular vectors w.r.t. non-zero singular values in $\hat{\mathbf{U}}$ following (Fang et al., 2024), which leads to an approximate null space and induces semantic degradation (see Fig. 2). To improve, we propose ***Prior Knowledge Refinement,*** a structured strategy for refining the retain set to enable accurate null-space construction, with three complementary techniques: (1) Influence-Based Prior Filtering (Sec. 4.1) to discard weakly affected non-target concepts to form a viable null space; (2) Directed Prior Augmentation (Sec. 4.2) to expand the retain set with targeted and semantically consistent variations; and (3) Invariant Equality Constraints (Sec. 4.3) to enforce equality constraints to preserve critical invariants during generation.

## 4.1 INFLUENCE-BASED PRIOR FILTERING (IPF)

Given a predefined retain set, existing editing-based methods (Gandikota et al., 2024; Gong et al., 2025) treat all non-target concepts equally when enforcing prior preservation. However, an overlooked fact is that parameter updates inherently induce output changes over non-target concepts, and these changes vary across different non-target concepts. This suggests that not all non-target concepts contribute equally to preserving prior knowledge, and weakly influenced concepts offer little benefit but introduce additional ranks that narrow the null space.

To this end, we propose an explicit and model-consistent metric (*i.e.*, ***prior shift***) to quantify how much a certain non-target concept is affected by concept erasure. Specifically, we isolate the effect of erasure by solving for a closed-form update $\mathbf{\Delta}_{\text{erase}}$ that minimizes only the erasure error $e_1$ while discarding the preservation term $e_0$ from Eq. 1:

$$\mathbf{\Delta}_{\text{erase}} = \arg\min_{\mathbf{\Delta}} \underbrace{\|(\mathbf{W}+\mathbf{\Delta})\mathbf{C}_1 - \mathbf{W}\mathbf{C}_*\|^2}_{e_1} + \underbrace{\|\mathbf{\Delta}\|^2}_{\text{regularization}} = \mathbf{W}\left(\mathbf{C}_*\mathbf{C}_1^\top - \mathbf{C}_1\mathbf{C}_1^\top\right)\left(\mathbf{I}+\mathbf{C}_1\mathbf{C}_1\right)^{-1},$$

(5)

where $\|\mathbf{\Delta}\|^2$ is introduced for convergence. For each non-target concept embedding $\mathbf{c}_0$, we define its prior shift as $\|\mathbf{\Delta}_{\text{erase}}\mathbf{c}_0\|^2$. This value offers a faithful reflection of how parameter updates perturb a non-target concept in the feature space with closed-form computation, and can naturally generalize to assessing multi-concept erasure effects. Based on this, we filter the original retain set $\mathbf{R}$ to focus only on highly influenced concepts:

$$\mathbf{R}_f : \mathbf{R} \mapsto \{\mathbf{c}_0 \in \mathbf{R} \mid \|\mathbf{\Delta}_{\text{erase}}\mathbf{c}_0\|^2 > \mu\},$$

(6)

where the mean value $\mu = \mathbb{E}_{\mathbf{c}_0 \sim \mathbf{R}}\left[\|\mathbf{\Delta}_{\text{erase}}\mathbf{c}_0\|^2\right]$ serves as a filtering threshold.

## 4.2 DIRECTED PRIOR AUGMENTATION (DPA)

To further enhance the prior preservation with broader retain coverage, an intuitive strategy is to augment the retain set by perturbing non-target embedding $\mathbf{c}_0$ with random noise (Lyu et al., 2024). However, this strategy would introduce meaningless embeddings that fail to generate semantically coherent images (*e.g.*, noise image), resulting in excessive preservation with increasing ranks. To search for more semantically consistent concepts, we introduce directed noise by projecting the random noise $\boldsymbol{\epsilon}$ onto the direction in which the model parameters $\mathbf{W}$ exhibit minimal variation. This operation ensures the perturbed embeddings express closer semantics to the original concept after being mapped by $\mathbf{W}$ (see Fig. 3). Specifically, we first derive a projection matrix $\mathbf{P}_{\text{min}}$:

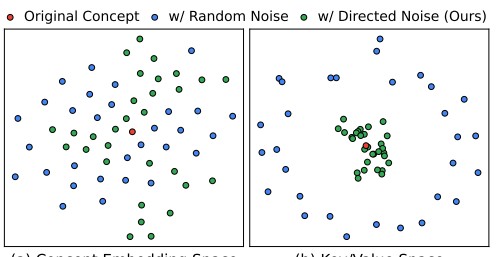

Figure 3: **t-SNE distribution of perturbing the original concept** with *random noise* and *directed noise.* (a) Similar to random noise, our method can span a broad concept embedding space. (b) Our directed noise preserves semantic similarity to the original concept with closer distances in the space mapped by $\mathbf{W}$.

$$\left\{\mathbf{U}_\mathbf{W}, \mathbf{\Lambda}_\mathbf{W}, \mathbf{U}_\mathbf{W}^\top\right\} = \text{SVD}\left(\mathbf{W}\right), \quad \mathbf{P}_{\text{min}} = \mathbf{U}_{\text{min}}\mathbf{U}_{\text{min}}^\top,$$

(7)

where $\mathbf{U}_{\text{min}} = \mathbf{U}_\mathbf{W}[:, -r :]$ denotes the singular vectors w.r.t. the smallest $r$ singular vectors[3], which represent the $r$ least-changing directions of $\mathbf{W}$ and constrain the rank of the augmented embeddings

---

[3]Empirically, the model parameter matrix $\mathbf{W}$ is usually full rank, thus its all singular values are non-zero.

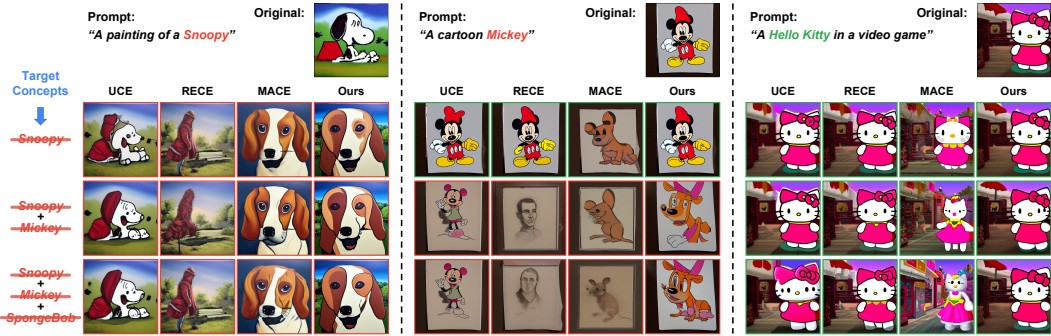

Figure 4: **Qualitative comparison of the few-concept erasure in erasing instances.** The erased and preserved generations are highlighted with **red** and **green** boxes, respectively. Our method exhibits consistent prior preservation with less semantic degradation for non-target concepts. For example, the middle column better retains details such as *Mickey*'s hat and button count, and the right column demonstrates more consistent *Hello Kitty* generations along with three concepts erased.

to a maximum of $r$. Then the directed noise $\boldsymbol{\epsilon} \cdot \mathbf{P}_{\min}$ is used to perturb the original embedding via:

$$\boldsymbol{c}_0' = \boldsymbol{c}_0 + \boldsymbol{\epsilon} \cdot \mathbf{P}_{\min}, \quad \boldsymbol{\epsilon} \sim \mathcal{N}(\mathbf{0}, \mathbf{I}). \tag{8}$$

Given a retain set $\mathbf{R}$, the augmentation process can be formulated as follows:

$$\mathbf{R}^{\text{aug}} : \mathbf{R} \mapsto \bigcup_{\boldsymbol{c}_0 \in \mathbf{R}} \left\{ \boldsymbol{c}_0 + \boldsymbol{\epsilon}_k \cdot \mathbf{P}_{\min} \mid k = 1, \dots, N_A \right\}, \tag{9}$$

where $N_A$ denotes the augmentation times and $\boldsymbol{c}_0 + \boldsymbol{\epsilon}_k \cdot \mathbf{P}_{\min}$ represents the $k$-th augmented embedding given $\boldsymbol{c}_0 \in \mathbf{R}$ using Eq. 8. In implementation, we first apply IPF to the original retain set $\mathbf{R}$ to obtain $\mathbf{R}_f$, and then apply DPA followed by IPF to $\mathbf{R}_f$ to produce $(\mathbf{R}_f)_f^{\text{aug}}$ as augmented concepts. Finally, we combine them to serve as the final refined retain set $\mathbf{R}_{\text{refine}} = \mathbf{R}_f \cup (\mathbf{R}_f)_f^{\text{aug}}$.

### 4.3 INVARIANT EQUALITY CONSTRAINTS (IEC)

In parallel, we identify certain invariants during the T2I generation process, *i.e.*, intermediate variables that remain unchanged under different sampling prompts. One such invariant is the CLIP-encoded `[SOT]` token. Since the encoding process is masked by causal attention and all prompts are prefixed with the fixed `[SOT]` token during tokenization, its embedding consistently remains unchanged during T2I process. Another invariant is the null-text embedding, as it corresponds to the unconditional generation under the classifier-free guidance (Ho & Salimans, 2022), which also remains unchanged despite prompt variations. Given the invariance of these embeddings, we consider additional preservation measures to ensure their outputs remain unchanged during concept erasure. Specifically, we introduce explicit equality constraints over invariants based on Eq. 3:

$$\min_{\boldsymbol{\Delta}} \underbrace{\|(\mathbf{W} + \boldsymbol{\Delta P})\mathbf{C}_1 - \mathbf{W}\mathbf{C}_*\|^2}_{e_1} + \underbrace{\|\boldsymbol{\Delta P}\|^2}_{\text{regularization}}, \quad \text{s.t.} \underbrace{(\boldsymbol{\Delta P})\mathbf{C}_2 = \mathbf{0}}_{\text{equality constraints}}, \tag{10}$$

where $\mathbf{C}_2$ stacks the invariant embeddings of `[SOT]` and null-text. Using the projection matrix $\mathbf{P}$ constructed from $\mathbf{R}_{\text{refine}}$, we can derive the closed-form solution of Eq. 10 using Lagrange Multipliers from Appx. B.3:

$$(\boldsymbol{\Delta P})_{\text{Ours}} = \mathbf{W}\left(\mathbf{C}_*\mathbf{C}_1^\top - \mathbf{C}_1\mathbf{C}_1^\top\right)\mathbf{P}\mathbf{Q}\mathbf{M}, \tag{11}$$

where

$$\mathbf{M} = \left(\mathbf{C}_1\mathbf{C}_1^\top\mathbf{P} + \mathbf{I}\right)^{-1}, \mathbf{Q} = \mathbf{I} - \mathbf{M}\mathbf{C}_2\left(\mathbf{C}_2^\top\mathbf{P}\mathbf{M}\mathbf{C}_2\right)^{-1}\mathbf{C}_2^\top\mathbf{P}. \tag{12}$$

This closed-form solution enforces the equality constraints by projecting the parameter update onto the subspace orthogonal to the invariant embeddings. Since image generation inevitably depends on these invariant embeddings, such constraints inherently preserve prior knowledge.

Table 1: **Quantitative comparison of the few-concept erasure** on instances (*left*) and artistic styles (*right*), where target concepts are highlighted in pink. Arrows indicate the preferred direction for each metric, and the best results are highlighted in **bold**. Our method consistently improves prior preservation on non-target and general concepts from MS-COCO while achieving effective concept erasure. Although our CS is not always the lowest for target concepts, Appx. D.1 shows our method is sufficient for erasure, and lower CS may further compromise prior preservation.

*Instances (left):*

| Concept | Snoopy | Mickey | Spongebob | Pikachu | Hello Kitty | MS-COCO | |
|---|---|---|---|---|---|---|---|
| | CS | CS | CS | CS | CS | CS | FID |
| SD v1.4 | 28.51 | 26.62 | 27.30 | 27.44 | 27.77 | 26.53 | - |
| *Erase Snoopy* | | | | | | | |
| | CS ↓ | FID ↓ | FID ↓ | FID ↓ | FID ↓ | CS ↑ | FID ↓ |
| ConAbl | 25.44 | 37.08 | 38.92 | 26.14 | 36.52 | 26.40 | 21.20 |
| MACE | 20.90 | 105.97 | 102.77 | 65.71 | 75.42 | 26.09 | 42.62 |
| RECE | **18.38** | 26.63 | 34.42 | 21.99 | 32.35 | 26.39 | 25.61 |
| UCE | 23.19 | 24.87 | 29.86 | 19.06 | 27.86 | 26.46 | 22.18 |
| Ours | 23.50 | **23.41** | **24.64** | **16.81** | **21.74** | **26.48** | **19.95** |
| *Erase Snoopy and Mickey* | | | | | | | |
| | CS ↓ | CS ↓ | FID ↓ | FID ↓ | FID ↓ | CS ↑ | FID ↓ |
| ConAbl | 25.26 | 26.58 | 45.08 | 35.57 | 41.48 | 26.42 | 24.34 |
| MACE | 20.53 | 20.63 | 112.01 | 91.72 | 106.88 | 25.50 | 55.15 |
| RECE | **18.57** | **19.14** | 35.85 | 26.05 | 40.77 | 26.31 | 30.30 |
| UCE | 23.60 | 24.79 | 30.58 | 23.51 | 31.76 | 26.38 | 26.06 |
| Ours | 23.58 | 23.62 | **29.67** | **22.51** | **28.23** | **26.47** | **23.66** |
| *Erase Snoopy and Mickey and Spongebob* | | | | | | | |
| | CS ↓ | CS ↓ | CS ↓ | FID ↓ | FID ↓ | CS ↑ | FID ↓ |
| ConAbl | 24.92 | 26.46 | 25.12 | 46.47 | 48.24 | 26.37 | 26.71 |
| MACE | 19.86 | 19.35 | 20.12 | 110.12 | 128.56 | 23.39 | 66.39 |
| RECE | **18.17** | **18.87** | **16.23** | 40.52 | 52.06 | 26.32 | 32.51 |
| UCE | 23.29 | 24.63 | 19.08 | 29.20 | 38.15 | 26.30 | 28.71 |
| Ours | 23.69 | 23.93 | 21.39 | **21.40** | **26.22** | **26.51** | **24.99** |

*Artistic styles (right):*

| Concept | Van Gogh | Picasso | Monet | Paul Gauguin | Caravaggio | MS-COCO | |
|---|---|---|---|---|---|---|---|
| | CS | CS | CS | CS | CS | CS | FID |
| SD v1.4 | 28.75 | 27.98 | 28.91 | 29.80 | 26.27 | 26.53 | - |
| *Erase Van Gogh* | | | | | | | |
| | CS ↓ | FID ↓ | FID ↓ | FID ↓ | FID ↓ | CS ↑ | FID ↓ |
| ConAbl | 28.16 | 77.01 | 63.80 | 63.20 | 79.25 | 26.46 | **18.36** |
| MACE | 26.66 | 69.92 | 60.88 | 56.18 | 69.04 | 26.50 | 23.15 |
| RECE | 26.39 | 60.57 | 61.09 | 47.07 | 72.85 | 26.52 | 23.54 |
| UCE | 28.10 | 43.02 | 40.49 | 32.62 | 61.72 | 26.54 | 19.63 |
| Ours | 26.29 | **35.86** | **16.85** | **24.94** | **39.75** | **26.55** | 20.36 |
| *Erase Picasso* | | | | | | | |
| | FID ↓ | CS ↓ | FID ↓ | FID ↓ | FID ↓ | CS ↑ | FID ↓ |
| ConAbl | 60.44 | 26.97 | 36.23 | 65.23 | 79.12 | 26.43 | 20.02 |
| MACE | 59.58 | 26.48 | 37.02 | 46.35 | 66.20 | 26.47 | 22.86 |
| RECE | 51.09 | 26.66 | 25.39 | 46.08 | 75.61 | 26.48 | 23.03 |
| UCE | 37.58 | 26.99 | **16.72** | 32.48 | 59.27 | 26.50 | 20.33 |
| Ours | **19.18** | **26.22** | 19.87 | **24.73** | **43.63** | **26.51** | **19.98** |
| *Erase Monet* | | | | | | | |
| | FID ↓ | FID ↓ | CS ↓ | FID ↓ | FID ↓ | CS ↑ | FID ↓ |
| ConAbl | 68.77 | 64.25 | 27.05 | 57.33 | 71.88 | 26.45 | 21.03 |
| MACE | 61.50 | 48.41 | 25.98 | 49.66 | 65.87 | 26.47 | 22.76 |
| RECE | 56.26 | 45.97 | 25.87 | 46.38 | 64.19 | 26.49 | 24.94 |
| UCE | 42.25 | **38.73** | 27.12 | 33.00 | 56.49 | **26.51** | 21.58 |
| Ours | **28.78** | 41.21 | **25.06** | **27.85** | **55.20** | 26.48 | **20.87** |

# 5 EXPERIMENTS

In this section, we conduct extensive experiments on three representative erasure tasks, including few-concept erasure, multi-concept erasure, and implicit concept erasure (Appx. D.4), validating our superior prior preservation. The compared baselines include ConAbl (Kumari et al., 2023), MACE (Lu et al., 2024a), RECE (Gong et al., 2025), and UCE (Gandikota et al., 2024), which have achieved SOTA performance across various concept erasure tasks. In implementation, we conduct all experiments on SDv1.4 (Stability AI, 2022) and generate each image using DPM-solver sampler (Lu et al., 2022) over 20 sampling steps with classifier-free guidance (Ho & Salimans, 2022) of 7.5. More implementation details and compared baselines can be found in Appx. C and Appx. D.3.

## 5.1 ON FEW-CONCEPT ERASURE

**Evaluation setup.** We evaluate few-concept erasure on instance erasure and artistic style erasure following (Lyu et al., 2024), using 80 instance templates and 30 artistic style templates with 10 images per template per concept. We use two metrics for evaluation: CLIP Score (CS) (Radford et al., 2021) for the text-image similarity and Fréchet Inception Distance (FID) (Heusel et al., 2017) for the distributional distance before and after erasure. Following (Lyu et al., 2024), we select non-target concepts with similar semantics to the target concept for comparison and report CS for targets and FID for non-targets in the main paper. Complete comparisons are presented in Appx. D.2. We further compare the generations on MS-COCO captions (Lin et al., 2014), where we generate images with the first 1,000 captions, and report CS and FID to measure general knowledge preservation.

**Analysis and discussion.** Table 1 compares the results of erasing various instance concepts and artistic styles. Our method consistently achieves the lowest FIDs across all non-target concepts, demonstrating superior prior preservation with minimal alteration to the original content. Moreover, we emphasize that our erasure is sufficiently effective, even without achieving the lowest CS, as shown in Figs. 4 and 7. On this basis, lower CS values typically indicate "over-erasure" of the target concept, since further reductions in CS after successful erasure usually come at the cost of prior preservation, as detailed in Appx. D.1. Notably, with the number of target concepts increasing from 1 to 3, our FID in *Pikachu* rises from 16.81 to 21.40 (4.59 ↑), while UCE increases from 19.06 to 29.20 (10.14 ↑). A similar pattern is observed in *Hello Kitty* (Our 4.48 ↑ *v.s.* UCE's 10.29 ↑), showing our superiority of prior preservation in erasing increasing target concepts.

Table 2: **Quantitative comparison of the multi-concept erasure** in erasing 10, 50, and 100 celebrities. The best results are highlighted in **bold**. Our method can effectively erase up to 100 celebrities simultaneously, achieving low $Acc_e$ (%) and high $Acc_r$ (%) that preserve non-target celebrities with minimal appearance changes. This yields the best overall erasure performance $H_o$ and competitive runtime (s), successfully erasing 100 concepts in just 5 seconds.

| | **Erase 10 Celebrities** | | | | **MS-COCO** | | **Erase 50 Celebrities** | | | | **MS-COCO** | | **Erase 100 Celebrities** | | | | **MS-COCO** | |
|---|---|---|---|---|---|---|---|---|---|---|---|---|---|---|---|---|---|---|
| | $Acc_e \downarrow$ | $Acc_r \uparrow$ | $H_o \uparrow$ | Time $\downarrow$ | CS $\uparrow$ | FID $\downarrow$ | $Acc_e \downarrow$ | $Acc_r \uparrow$ | $H_o \uparrow$ | Time $\downarrow$ | CS $\uparrow$ | FID $\downarrow$ | $Acc_e \downarrow$ | $Acc_r \uparrow$ | $H_o \uparrow$ | Time $\downarrow$ | CS $\uparrow$ | FID $\downarrow$ |
| SD v1.4 | 91.99 | 89.66 | 14.70 | - | 26.53 | - | 93.08 | 89.66 | 12.85 | - | 26.53 | - | 90.18 | 89.66 | 17.70 | - | 26.53 | - |
| ConAbl | 60.76 | 77.89 | 52.19 | 900 | 25.60 | 42.12 | 64.00 | 75.44 | 48.74 | 4,500 | 14.30 | 255.36 | 42.86 | 58.82 | 57.97 | 9,000 | 14.93 | 235.27 |
| UCE | 0.20 | 71.19 | 83.10 | 1.5 | 24.07 | 83.81 | 0.00 | 31.94 | 48.41 | 1.8 | 13.45 | 209.93 | 0.00 | 20.92 | 34.60 | 2.1 | 13.49 | 185.46 |
| RECE | 0.34 | 67.43 | 80.44 | 2.5 | 16.75 | 170.65 | 1.03 | 19.77 | 32.95 | 6.3 | 13.49 | 213.39 | 2.43 | 23.71 | 38.16 | 11.0 | 12.09 | 177.57 |
| MACE | 1.62 | 87.73 | 92.75 | 207 | 26.36 | 37.25 | 3.41 | 84.31 | 90.03 | 936 | 25.45 | 45.31 | 4.80 | 80.20 | 87.06 | 1736 | 24.80 | 50.41 |
| Ours | 1.81 | 89.09 | **93.42** | 3.8 | **26.47** | **30.02** | 3.46 | 88.48 | **92.34** | 4.2 | **26.46** | **39.23** | 5.87 | 85.54 | **89.63** | 5.0 | **26.22** | **44.97** |

## 5.2 ON MULTI-CONCEPT ERASURE

**Evaluation setup.** Another more realistic erasure scenario is multi-concept erasure, where massive concepts are required to be erased at once. We follow the experiment setup in Lu et al. (2024a) for erasing multiple celebrities, where we experiment with erasing 10, 50, and 100 celebrities and collect another 100 celebrities as non-target concepts. Specifically, we prepare 5 prompt templates for each celebrity concept. For non-target concepts, we generate 1 image per template for each of the 100 concepts, totaling 500 images. For target concepts, we adjust the per-concept quantity to maintain a total of 500 images (*e.g.*, erasing 10 celebrities involves generating 10 images with 5 templates per concept). In evaluation, we adopt GIPHY Celebrity Detector (GCD) (Hasty et al., 2019) and measure the top-1 GCD accuracy, indicated by $Acc_e$ for erased target concepts and $Acc_r$ for retained non-target concepts. Meanwhile, the harmonic mean $H_o = \frac{2}{(1-Acc_e)^{-1}+(Acc_r)^{-1}}$ is adopted to assess the overall erasure performance. Additionally, we report the results on MS-COCO to demonstrate the prior preservation of general concepts.

**Analysis and discussion.** Table 2 showcases a notable improvement of our method on multi-concept erasure, particularly in prior preservation with the highest $Acc_r$. In comparison with the SOTA method, MACE (Lu et al., 2024a), our method achieves superior prior preservation with better $Acc_r$, while maintaining comparable erasure efficacy, as reflected in similar $Acc_e$, resulting in the best overall erasure performance indicated by the highest $H_o$. Meanwhile, our method attains the lowest FID across all methods on MS-COCO. The other methods, UCE (Gandikota et al., 2024) and RECE (Gong et al., 2025), although achieving considerable balance in few-concept erasure, fail to maintain this balance as the number of target concepts

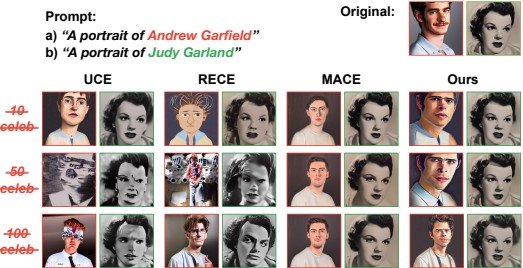

Figure 5: **Quantitative comparison of the multi-concept erasure** in erasing celebrities (*celeb*). The erased and preserved generations are marked with **red** and **green** boxes. Our method precisely erases 100 celebrities while preserving generations of other non-target concepts.

increases as shown in Fig. 5, with catastrophic prior damage evidenced by MS-COCO as well. Notably, our method can erase up to 100 celebrities in 5 seconds, whereas MACE requires around 30 minutes ($\times 350$ time). In real-world scenarios, this efficiency underscores our potential for the instant erasure of massive concepts.

## 5.3 FURTHER ANALYSIS

**More applications on other T2I models.** To validate the transferability of our method across versatile applications, we conduct further experiments on various T2I models with different weights and architectures, including: (1) Composite concept erasure on DreamShaper (Lykon, 2023) and RealisticVision (SG161222, 2023) from Fig 6 (a): Our method can precisely erase the target concept(s) while preserving other non-target elements within the prompt, such as the *Van Gogh*-style background (2nd column) and the *Snoopy* character (3rd column). (2) Knowledge editing on SDXL (Podell et al., 2023) from Fig 6 (b): The arbitrary nature of anchor concepts allows us to edit the pre-trained model knowledge. Herein, our method effectively edits the model knowledge while

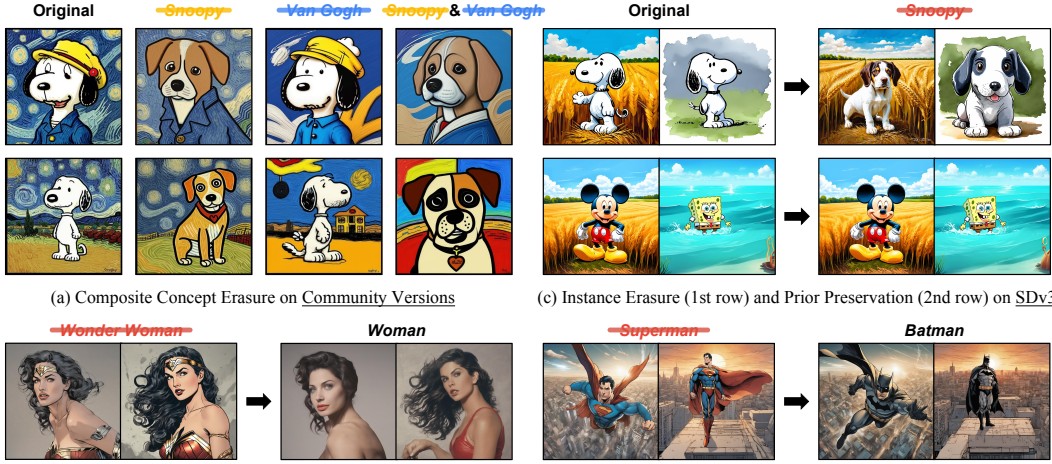

(a) Composite Concept Erasure on Community Versions

(c) Instance Erasure (1st row) and Prior Preservation (2nd row) on SDv3

(b) Knowledge Editing (*e.g.*, "*Wonder Woman → Woman*" and "*Superman → Batman*") on SDXL

Figure 6: **More applications across various T2I diffusion models.** (a) We conduct composite concept erasure for "*Snoopy + Van Gogh*" on DreamShaper (Lykon, 2023) (1st row) and RealisticVision (SG161222, 2023) (2nd row). (b) Our method also enables model knowledge editing by specifying the anchor concept on SDXL (Podell et al., 2023). (c) Our method can seamlessly transfer to novel DiT-based T2I models, *e.g.*, SDv3 (Esser et al., 2024).

maintaining the overall layout and semantics of the generated images. (3) Instance erasure on SDv3 (Esser et al., 2024) from Fig 6 (c): To accommodate the diffusion transformer (DiT) (Peebles & Xie, 2023) architecture in T2I models, we adapt our method to a DiT-based model, demonstrating a well-balanced trade-off between erasure (1st row) and preservation (2nd row) as well.

**Component ablation.** From Table 3, we compare the individual impact of our components on prior preservation and draw the following conclusions: (1) Impact of IEC (Ablation 1 *v.s.* 2): IEC reduces the non-target FID and the MS-COCO FID, demonstrating its effectiveness by preserving invariant embeddings with equality constraints. (2) Impact of IPF (Ablation 2 *v.s.* 3): Incorporating IPF results in a significant improvement in both FIDs, underscoring its critical role in filtering out less-influenced concepts in the retain set to mitigate semantic degradation. (3) Impact of DPA (Ablation 4 *v.s.* Ours): DPA improves RPA with directed noise and leads to a substantial improvement in non-target and MS-COCO FIDs, highlighting its advantage by introducing semantically similar con-

Table 3: **Ablation study** on proposed components in erasing *Van Gogh*, with the non-target FID averaged over the other four artistic styles from Table 1. Ablation 1 corresponds to the original objective from (Fang et al., 2024) in Eq. 3. The ablated components include: IEC (Invariant Equality Constraints), IPF (Influence-based Prior Filtering), RPA (Random Prior Augmentation), and DPA (Directed Prior Augmentation).

| Ablation | Components | | | | Target | Non-Target | MS-COCO | |
|---|---|---|---|---|---|---|---|---|
| | IEC | IPF | RPA | DPA | CS ↓ | FID ↓ | CS ↑ | FID ↓ |
| 1 | ✗ | ✗ | ✗ | ✗ | 27.20 | 50.43 | 26.42 | 26.33 |
| 2 | ✓ | ✗ | ✗ | ✗ | 27.20 | 48.17 | 26.44 | 24.95 |
| 3 | ✓ | ✓ | ✗ | ✗ | 26.68 | 38.02 | 26.54 | 20.57 |
| 4 | ✓ | ✓ | ✓ | ✗ | 26.30 | 32.62 | 26.52 | 20.99 |
| Ours | ✓ | ✓ | ✗ | ✓ | **26.29** | **29.35** | **26.55** | **20.36** |
| SD v1.4 | - | - | - | - | 28.75 | - | 26.53 | - |

cepts into the refined retain set. To conclude, the proposed three components (*i.e.*, IEC, IPF, and DPA) improve the prior preservation from different perspectives and contribute to our method with the best prior preservation under null space constraints. More ablations are presented in Appx. D.5.

# 6 CONCLUSION

This paper introduced SPEED, a scalable, precise, and efficient concept erasure method for T2I diffusion models. It formulates concept erasure as a null-space constrained optimization problem, facilitating effective prior preservation along with precise erasure efficacy. Critically, SPEED overcomes the inefficacy of editing-based methods in multi-concept erasure while circumventing the prohibitive computational costs associated with training-based approaches. With our proposed Prior Knowledge Refinement involving three complementary techniques, SPEED not only ensures superior prior preservation but also achieves a $350\times$ acceleration in multi-concept erasure, establishing itself as a scalable and practical solution for real-world applications.

ETHICS STATEMENT

This work introduces a method for concept erasure in text-to-image diffusion models to address ethical concerns such as copyright infringement, privacy violations, and the generation of offensive content. By precisely removing specific target concepts while preserving the quality and semantics of non-target outputs, the proposed approach enhances the safety, reliability, and controllability of generative models. The method operates through parameter-space editing without requiring access to private data or involving human subjects, ensuring ethical integrity throughout the research process and promoting responsible deployment of generative AI technologies.

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

## A PRELIMINARIES

**T2I diffusion models.** T2I generation has seen significant advancements with diffusion models, particularly Latent Diffusion Models (LDMs) (Rombach et al., 2022). Unlike pixel-space diffusion, LDMs operate in the latent space of a pretrained autoencoder, reducing computational costs while maintaining high-quality synthesis. LDMs consist of a vector-quantized autoencoder (Van Den Oord et al., 2017; Esser et al., 2021) and a diffusion model (Dhariwal & Nichol, 2021; Ho et al., 2020; Sohl-Dickstein et al., 2015; Kingma et al., 2021; Song et al., 2020b; Wang et al., 2024c; 2026; Li et al., 2025). The autoencoder encodes an image $x$ into a latent representation $z = \mathcal{E}(x)$ and reconstructs it via $x \approx \mathcal{D}(z)$. The diffusion model learns to generate latent codes through a denoising process. The training objective is given by (Ho et al., 2020; Rombach et al., 2022):

$$\mathcal{L}_{\text{LDM}} = \mathbb{E}_{z \sim \mathcal{E}(x), c, \epsilon \sim \mathcal{N}(0,1), t} \left[ \| \epsilon - \epsilon_\theta(z_t, t, c) \|_2^2 \right], \tag{13}$$

where $z_t$ is the noisy latent at timestep $t$, $\epsilon$ is Gaussian noise, $\epsilon_\theta$ is the denoising network, and $c$ is conditioning information from text, class labels, or segmentation masks (Rombach et al., 2022). During inference, a latent $z_T$ is sampled from a Gaussian prior and progressively denoised to obtain $z_0$, which is then decoded into an image via $x_0 \approx \mathcal{D}(z_0)$.

**Cross-attention mechanisms.** Current T2I diffusion models usually leverage a generative framework to synthesize images conditioned on textual descriptions in the latent space (Rombach et al., 2022). The conditioning mechanism is implemented through cross-attention (CA) layers. Specifically, textual descriptions are first tokenized into $n$ tokens and embedded into a sequence of vectors $e \in \mathbb{R}^{d_0 \times n}$ via a pre-trained CLIP model (Radford et al., 2021). These text embeddings serve as the key $\mathbf{K} \in \mathbb{R}^{n \times d_k}$ and value $\mathbf{V} \in \mathbb{R}^{n \times d_v}$ inputs using parametric projection matrices $\mathbf{W_K} \in \mathbb{R}^{d_k \times d_0}$ and $\mathbf{W_V} \in \mathbb{R}^{d_v \times d_0}$, while the intermediate image representations act as the query $\mathbf{Q} \in \mathbb{R}^{m \times d_k}$. The cross-attention mechanism is defined as:

$$\text{Attention}(\mathbf{Q}, \mathbf{K}, \mathbf{V}) = \text{softmax}\left( \frac{\mathbf{Q}\mathbf{K}^T}{\sqrt{d_k}} \right) \mathbf{V}. \tag{14}$$

This alignment enables the model to capture semantic correlations between the textual input and the visual features, ensuring that the generated images are semantically consistent with the provided text prompts.

## B PROOF AND DERIVATION

### B.1 DERIVING THE CLOSED-FORM SOLUTION FOR UCE

From Eq. 1, we are tasked with minimizing the following editing objective, where the hyperparameters $\alpha$ and $\beta$ correspond to the weights of the erasure error $e_1$ and the preservation error $e_0$, respectively:

$$\min_{\mathbf{\Delta}} \left[ \alpha \| (\mathbf{W} + \mathbf{\Delta})\mathbf{C}_1 - \mathbf{W}\mathbf{C}_* \|^2 + \beta \| \mathbf{\Delta}\mathbf{C}_0 \|^2 \right]. \tag{15}$$

To derive the closed-form solution, we begin by computing the gradient of the objective function with respect to $\mathbf{\Delta}$. The gradient is given by:

$$\alpha \left( \mathbf{W}\mathbf{C}_1 - \mathbf{W}\mathbf{C}_* + \mathbf{\Delta}\mathbf{C}_1 \right) \mathbf{C}_1^\top + \beta \mathbf{\Delta}\mathbf{C}_0\mathbf{C}_0^\top = 0. \tag{16}$$

Solving the resulting equation yields the closed-form solution for $\mathbf{\Delta}_{\text{UCE}}$:

$$\mathbf{\Delta}_{\text{UCE}} = \alpha\mathbf{W} \left( \mathbf{C}_*\mathbf{C}_1^\top - \mathbf{C}_1\mathbf{C}_1^\top \right) \left( \alpha\mathbf{C}_1\mathbf{C}_1^\top + \beta\mathbf{C}_0\mathbf{C}_0^\top \right)^{-1}. \tag{17}$$

In practice, an additional identity matrix $\mathbf{I}$ with hyperparameter $\lambda$ is added to $\left( \alpha\mathbf{C}_1\mathbf{C}_1^\top + \beta\mathbf{C}_0\mathbf{C}_0^\top \right)^{-1}$ to ensure its invertibility. This modification results in the following closed-form solution for UCE:

$$\mathbf{\Delta}_{\text{UCE}} = \alpha\mathbf{W} \left( \mathbf{C}_*\mathbf{C}_1^\top - \mathbf{C}_1\mathbf{C}_1^\top \right) \left( \alpha\mathbf{C}_1\mathbf{C}_1^\top + \beta\mathbf{C}_0\mathbf{C}_0^\top + \lambda\mathbf{I} \right)^{-1}. \tag{18}$$

### B.2 Proof of the Lower Bound of $e_0$ for UCE

Herein, we aim to establish the existence of a strictly positive constant $c > 0$ such that

$$e_0 = \|\mathbf{\Delta}_{\text{UCE}}\mathbf{C}_0\|^2 = \|\alpha\mathbf{W}\left(\mathbf{C}_*\mathbf{C}_1^\top - \mathbf{C}_1\mathbf{C}_1^\top\right)\left(\alpha\mathbf{C}_1\mathbf{C}_1^\top + \beta\mathbf{C}_0\mathbf{C}_0^\top + \lambda\mathbf{I}\right)^{-1}\mathbf{C}_0\|^2 \geq c > 0. \quad (19)$$

**Assumption B.1.** *We assume that $\alpha, \beta, \lambda \neq 0$, that $\mathbf{W}$ is a full-rank matrix, and that $\mathbf{C}_0\mathbf{C}_0^\top$ is rank-deficient. Furthermore, we assume that*

$$\mathbf{C}_*\mathbf{C}_1^\top - \mathbf{C}_1\mathbf{C}_1^\top \neq \mathbf{0}.$$

*Proof.* Define the matrix $\mathbf{M}$ as

$$\mathbf{M} = \alpha\mathbf{C}_1\mathbf{C}_1^\top + \beta\mathbf{C}_0\mathbf{C}_0^\top + \lambda\mathbf{I}. \quad (20)$$

Since $\lambda > 0$ and $\mathbf{I}$ is positive definite, it follows that $\mathbf{M}$ is strictly positive definite and therefore invertible.

Rewriting $e_0$ by defining $\mathbf{B} = \mathbf{M}^{-1}\mathbf{C}_0$, we obtain

$$e_0 = \|\alpha\mathbf{W}(\mathbf{C}_*\mathbf{C}_1^\top - \mathbf{C}_1\mathbf{C}_1^\top)\mathbf{B}\|^2. \quad (21)$$

Applying the singular value bound for matrix products, we have

$$\|\mathbf{X}\mathbf{Y}\| \geq \sigma_{\min}(\mathbf{X})\|\mathbf{Y}\|, \quad (22)$$

where $\sigma_{\min}(\mathbf{X})$ is the smallest singular value of $\mathbf{X}$. Applying this inequality, we obtain

$$\|\mathbf{W}(\mathbf{C}_*\mathbf{C}_1^\top - \mathbf{C}_1\mathbf{C}_1^\top)\mathbf{B}\| \geq \sigma_{\min}(\mathbf{W})\|(\mathbf{C}_*\mathbf{C}_1^\top - \mathbf{C}_1\mathbf{C}_1^\top)\mathbf{B}\|. \quad (23)$$

We start with the singular value decomposition (SVD) of the matrix $\mathbf{C}_*\mathbf{C}_1^\top - \mathbf{C}_1\mathbf{C}_1^\top$, given by

$$\mathbf{C}_*\mathbf{C}_1^\top - \mathbf{C}_1\mathbf{C}_1^\top = \mathbf{U}\mathbf{\Sigma}\mathbf{V}^\top. \quad (24)$$

Here, $\mathbf{U}$ and $\mathbf{V}$ are orthogonal matrices, and

$$\mathbf{\Sigma} = \text{diag}(\sigma_1, \sigma_2, \ldots, \sigma_r, 0, \ldots, 0) \quad (25)$$

is a diagonal matrix containing the singular values $\sigma_1 \geq \sigma_2 \geq \cdots \geq \sigma_r > 0$, followed by zeros.

Multiplying both sides by $\mathbf{B}$, we obtain

$$(\mathbf{C}_*\mathbf{C}_1^\top - \mathbf{C}_1\mathbf{C}_1^\top)\mathbf{B} = \mathbf{U}\mathbf{\Sigma}\mathbf{V}^\top\mathbf{B}. \quad (26)$$

Define the projection of $\mathbf{B}$ onto the subspace spanned by the right singular vectors as

$$\mathbf{B}_{\text{proj}} = \mathbf{V}^\top\mathbf{B}. \quad (27)$$

Then, we can rewrite the expression as

$$(\mathbf{C}_*\mathbf{C}_1^\top - \mathbf{C}_1\mathbf{C}_1^\top)\mathbf{B} = \mathbf{U}\mathbf{\Sigma}\mathbf{B}_{\text{proj}}. \quad (28)$$

Taking norms on both sides and using the fact that orthogonal transformations preserve norms, we get

$$\|(\mathbf{C}_*\mathbf{C}_1^\top - \mathbf{C}_1\mathbf{C}_1^\top)\mathbf{B}\| = \|\mathbf{\Sigma}\mathbf{B}_{\text{proj}}\|. \quad (29)$$

Since $\mathbf{\Sigma}$ is a diagonal matrix, its smallest nonzero singular value $\sigma_r$ provides a lower bound:

$$\|\mathbf{\Sigma}\mathbf{B}_{\text{proj}}\| \geq \sigma_r\|\mathbf{B}_{\text{proj}}\|. \quad (30)$$

Next, we establish a lower bound for $\|\mathbf{B}_{\text{proj}}\|$. Given that $\mathbf{V}$ is composed of right singular vectors, there exists a smallest non-zero singular value $c_1$ such that:

$$\|\mathbf{B}_{\text{proj}}\| \geq c_1\|\mathbf{B}\|. \quad (31)$$

Combining these inequalities, we obtain

$$\|(\mathbf{C}_*\mathbf{C}_1^\top - \mathbf{C}_1\mathbf{C}_1^\top)\mathbf{B}\| \geq \sigma_r\|\mathbf{B}_{\text{proj}}\| \geq \sigma_r c_1\|\mathbf{B}\|. \quad (32)$$

Since $\mathbf{M}$ is positive definite, we use the standard norm inequality for an invertible matrix $\mathbf{M}$, which states that for any matrix $\mathbf{X}$,

$$\|\mathbf{MX}\| \leq \|\mathbf{M}\|\|\mathbf{X}\|. \tag{33}$$

Setting $\mathbf{X} = \mathbf{M}^{-1}\mathbf{C}_0$, we obtain

$$\|\mathbf{MM}^{-1}\mathbf{C}_0\| \leq \|\mathbf{M}\|\|\mathbf{M}^{-1}\mathbf{C}_0\|. \tag{34}$$

Since $\mathbf{MM}^{-1} = \mathbf{I}$, the left-hand side simplifies to $\|\mathbf{C}_0\|$, yielding

$$\|\mathbf{C}_0\| \leq \|\mathbf{M}\|\|\mathbf{M}^{-1}\mathbf{C}_0\|. \tag{35}$$

Dividing both sides by $\|\mathbf{M}\|$, we obtain

$$\|\mathbf{M}^{-1}\mathbf{C}_0\| \geq \frac{1}{\|\mathbf{M}\|}\|\mathbf{C}_0\|. \tag{36}$$

Thus, it follows that

$$\|\mathbf{B}\| = \|\mathbf{M}^{-1}\mathbf{C}_0\| \geq \frac{1}{\|\mathbf{M}\|}\|\mathbf{C}_0\|. \tag{37}$$

Combining the above results, we obtain

$$\|\mathbf{W}(\mathbf{C}_*\mathbf{C}_1^\top - \mathbf{C}_1\mathbf{C}_1^\top)\mathbf{B}\| \geq \sigma_{\min}(\mathbf{W})\sigma_r c_1 \frac{1}{\|\mathbf{M}\|}\|\mathbf{C}_0\|. \tag{38}$$

Squaring both sides, we conclude that

$$e_0 = \|\alpha\mathbf{W}(\mathbf{C}_*\mathbf{C}_1^\top - \mathbf{C}_1\mathbf{C}_1^\top)\mathbf{B}\|^2 \geq \alpha^2\sigma_{\min}^2(\mathbf{W})\sigma_r^2 c_1^2 \frac{1}{\|\mathbf{M}\|^2}\|\mathbf{C}_0\|^2. \tag{39}$$

Since all terms on the right-hand side are strictly positive by assumption, we establish the existence of a positive lower bound $c > 0$ such that

$$e_0 \geq c > 0. \tag{40}$$

This completes the proof. $\square$

### B.3 DERIVING THE CLOSED-FORM SOLUTION FOR SPEED

From Eq. 10, we are tasked with minimizing the following editing objective:

$$\min_{\mathbf{\Delta}} \|(\mathbf{W} + \mathbf{\Delta P})\mathbf{C}_1 - \mathbf{WC}_*\|^2 + \|\mathbf{\Delta P}\|^2, \quad \text{s.t. } (\mathbf{\Delta P})\mathbf{C}_2 = \mathbf{0}. \tag{41}$$

This is a weighted least squares problem subject to an equality constraint. To solve it, we first formulate the Lagrangian function, where $\mathbf{\Lambda}$ is the Lagrange multiplier:

$$\mathcal{L}(\mathbf{\Delta}, \mathbf{\Lambda}) = \|(\mathbf{W} + \mathbf{\Delta P})\mathbf{C}_1 - \mathbf{WC}_*\|^2 + \|\mathbf{\Delta P}\|^2 + \mathbf{\Lambda}^\top ((\mathbf{\Delta P})\mathbf{C}_2). \tag{42}$$

We compute the gradient of the Lagrangian function in Eq. 42 with respect to $\mathbf{\Delta}$ and set it to zero, yielding the following equation for $\mathbf{\Delta}$:

$$\frac{\partial\mathcal{L}(\mathbf{\Delta}, \mathbf{\Lambda})}{\partial\mathbf{\Delta}} = 2\left((\mathbf{W} + \mathbf{\Delta P})\mathbf{C}_1 - \mathbf{WC}_*\right)\mathbf{C}_1^\top\mathbf{P}^\top + 2\mathbf{\Delta P}\mathbf{P}^\top + \mathbf{\Lambda}\mathbf{C}_2^\top\mathbf{P}^\top = \mathbf{0}. \tag{43}$$

Given that the projection matrix $\mathbf{P}$ is derived from $R_{\text{refine}}$ using Eq. 2, $\mathbf{P}$ is a symmetric matrix (*i.e.*, $\mathbf{P} = \mathbf{P}^\top$) and an idempotent matrix (*i.e.*, $\mathbf{P}^2 = \mathbf{P}$), the above formulation can be simplified to:

$$\frac{\partial\mathcal{L}(\mathbf{\Delta}, \mathbf{\Lambda})}{\partial\mathbf{\Delta}} = 2\left((\mathbf{W} + \mathbf{\Delta P})\mathbf{C}_1 - \mathbf{WC}_*\right)\mathbf{C}_1^\top\mathbf{P} + 2\mathbf{\Delta P} + \mathbf{\Lambda}\mathbf{C}_2^\top\mathbf{P} = \mathbf{0}. \tag{44}$$

Therefore, we can obtain the closed-form solution for $\mathbf{\Delta P}$ from this equation:

$$\mathbf{\Delta P} = (\mathbf{WC}_*\mathbf{C}_1^\top\mathbf{P} - \mathbf{WC}_1\mathbf{C}_1^\top\mathbf{P} - \frac{1}{2}\mathbf{\Lambda}\mathbf{C}_2^\top\mathbf{P})(\mathbf{C}_1\mathbf{C}_1^\top\mathbf{P} + \mathbf{I})^{-1}. \tag{45}$$

Table 4: **Evaluation setup for multi-concept erasure.** This dataset contains an erasure set with 100 celebrities and a retain set with another 100 celebrities. We experiment with erasing 10, 50, and 100 celebrities with the predefined target concepts and the entire retain set is utilized in all cases.

| Group | Number | Anchor Concept | Celebrity |
|---|---|---|---|
| Erasure Set | 10 | 'person' | 'Adam Driver', 'Adriana Lima', 'Amber Heard', 'Amy Adams', 'Andrew Garfield', 'Angelina Jolie', 'Anjelica Huston', 'Anna Faris', 'Anna Kendrick', 'Anne Hathaway' |
| | 50 | 'person' | 'Adam Driver', 'Adriana Lima', 'Amber Heard', 'Amy Adams', 'Andrew Garfield', 'Angelina Jolie', 'Anjelica Huston', 'Anna Faris', 'Anna Kendrick', 'Anne Hathaway', 'Arnold Schwarzenegger', 'Barack Obama', 'Beth Behrs', 'Bill Clinton', 'Bob Dylan', 'Bob Marley', 'Bradley Cooper', 'Bruce Willis', 'Bryan Cranston', 'Cameron Diaz', 'Channing Tatum', 'Charlie Sheen', 'Charlize Theron', 'Chris Evans', 'Chris Hemsworth', 'Chris Pine', 'Chuck Norris', 'Courteney Cox', 'Demi Lovato', 'Drake', 'Drew Barrymore', 'Dwayne Johnson', 'Ed Sheeran', 'Elon Musk', 'Elvis Presley', 'Emma Stone', 'Frida Kahlo', 'George Clooney', 'Glenn Close', 'Gwyneth Paltrow', 'Harrison Ford', 'Hillary Clinton', 'Hugh Jackman', 'Idris Elba', 'Jake Gyllenhaal', 'James Franco', 'Jared Leto', 'Jason Momoa', 'Jennifer Aniston', 'Jennifer Lawrence' |
| | 100 | 'person' | 'Adam Driver', 'Adriana Lima', 'Amber Heard', 'Amy Adams', 'Andrew Garfield', 'Angelina Jolie', 'Anjelica Huston', 'Anna Faris', 'Anna Kendrick', 'Anne Hathaway', 'Arnold Schwarzenegger', 'Barack Obama', 'Beth Behrs', 'Bill Clinton', 'Bob Dylan', 'Bob Marley', 'Bradley Cooper', 'Bruce Willis', 'Bryan Cranston', 'Cameron Diaz', 'Channing Tatum', 'Charlie Sheen', 'Charlize Theron', 'Chris Evans', 'Chris Hemsworth', 'Chris Pine', 'Chuck Norris', 'Courteney Cox', 'Demi Lovato', 'Drake', 'Drew Barrymore', 'Dwayne Johnson', 'Ed Sheeran', 'Elon Musk', 'Elvis Presley', 'Emma Stone', 'Frida Kahlo', 'George Clooney', 'Glenn Close', 'Gwyneth Paltrow', 'Harrison Ford', 'Hillary Clinton', 'Hugh Jackman', 'Idris Elba', 'Jake Gyllenhaal', 'James Franco', 'Jared Leto', 'Jason Momoa', 'Jennifer Aniston', 'Jennifer Lawrence', 'Jennifer Lopez', 'Jeremy Renner', 'Jessica Biel', 'Jessica Chastain', 'John Oliver', 'John Wayne', 'Johnny Depp', 'Julianne Hough', 'Justin Timberlake', 'Kate Bosworth', 'Kate Winslet', 'Leonardo Dicaprio', 'Margot Robbie', 'Mariah Carey', 'Melania Trump', 'Meryl Streep', 'Mick Jagger', 'Mila Kunis', 'Milla Jovovich', 'Morgan Freeman', 'Nick Jonas', 'Nicolas Cage', 'Nicole Kidman', 'Octavia Spencer', 'Olivia Wilde', 'Oprah Winfrey', 'Paul Mccartney', 'Paul Walker', 'Peter Dinklage', 'Philip Seymour Hoffman', 'Reese Witherspoon', 'Richard Gere', 'Ricky Gervais', 'Rihanna', 'Robin Williams', 'Ronald Reagan', 'Ryan Gosling', 'Ryan Reynolds', 'Shia Labeouf', 'Shirley Temple', 'Spike Lee', 'Stan Lee', 'Theresa May', 'Tom Cruise', 'Tom Hanks', 'Tom Hardy', 'Tom Hiddleston', 'Whoopi Goldberg', 'Zac Efron', 'Zayn Malik' |
| Retain Set | 10, 50, and 100 | - | 'Aaron Paul', 'Alec Baldwin', 'Amanda Seyfried', 'Amy Poehler', 'Amy Schumer', 'Amy Winehouse', 'Andy Samberg', 'Aretha Franklin', 'Avril Lavigne', 'Aziz Ansari', 'Barry Manilow', 'Ben Affleck', 'Ben Stiller', 'Benicio Del Toro', 'Bette Midler', 'Betty White', 'Bill Murray', 'Bill Nye', 'Britney Spears', 'Brittany Snow', 'Bruce Lee', 'Burt Reynolds', 'Charles Manson', 'Christie Brinkley', 'Christina Hendricks', 'Clint Eastwood', 'Countess Vaughn', 'Dakota Johnson', 'Dane Dehaan', 'David Bowie', 'David Tennant', 'Denise Richards', 'Doris Day', 'Dr Dre', 'Elizabeth Taylor', 'Emma Roberts', 'Fred Rogers', 'Gal Gadot', 'George Bush', 'George Takei', 'Gillian Anderson', 'Gordon Ramsey', 'Halle Berry', 'Harry Dean Stanton', 'Harry Styles', 'Hayley Atwell', 'Heath Ledger', 'Henry Cavill', 'Jackie Chan', 'Jada Pinkett Smith', 'James Garner', 'Jason Statham', 'Jeff Bridges', 'Jennifer Connelly', 'Jensen Ackles', 'Jim Morrison', 'Jimmy Carter', 'Joan Rivers', 'John Lennon', 'Johnny Cash', 'Jon Hamm', 'Judy Garland', 'Julianne Moore', 'Justin Bieber', 'Kaley Cuoco', 'Kate Upton', 'Keanu Reeves', 'Kim Jong Un', 'Kirsten Dunst', 'Kristen Stewart', 'Krysten Ritter', 'Lana Del Rey', 'Leslie Jones', 'Lily Collins', 'Lindsay Lohan', 'Liv Tyler', 'Lizzy Caplan', 'Maggie Gyllenhaal', 'Matt Damon', 'Matt Smith', 'Matthew Mcconaughey', 'Maya Angelou', 'Megan Fox', 'Mel Gibson', 'Melanie Griffith', 'Michael Cera', 'Michael Ealy', 'Natalie Portman', 'Neil Degrasse Tyson', 'Niall Horan', 'Patrick Stewart', 'Paul Rudd', 'Paul Wesley', 'Pierce Brosnan', 'Prince', 'Queen Elizabeth', 'Rachel Dratch', 'Rachel Mcadams', 'Reba Mcentire', 'Robert De Niro' |

Next, we differentiate the Lagrangian function in Eq. 42 with respect to $\mathbf{\Lambda}$ and set it to zero:

$$\frac{\partial \mathcal{L}(\mathbf{\Delta}, \mathbf{\Lambda})}{\partial \mathbf{\Lambda}} = (\mathbf{\Delta P})\mathbf{C}_2 = \mathbf{0}. \tag{46}$$

For simplicity, we define $\mathbf{M} = (\mathbf{C}_1\mathbf{C}_1^\top \mathbf{P} + \mathbf{I})^{-1}$. Then, we substitute the result of Eq. 45 into Eq. 46 and obtain:

$$(\mathbf{W}\mathbf{C}_*\mathbf{C}_1^\top \mathbf{P} - \mathbf{W}\mathbf{C}_1\mathbf{C}_1^\top \mathbf{P} - \frac{1}{2}\mathbf{\Lambda}\mathbf{C}_2^\top \mathbf{P})\mathbf{M}\mathbf{C}_2 = \mathbf{0}. \tag{47}$$

Solving this equation leads to:

$$\frac{1}{2}\mathbf{\Lambda} = \mathbf{W}(\mathbf{C}_*\mathbf{C}_1^\top - \mathbf{C}_1\mathbf{C}_1^\top)\mathbf{P}\mathbf{M}\mathbf{C}_2(\mathbf{C}_2^\top \mathbf{P}\mathbf{M}\mathbf{C}_2)^{-1}. \tag{48}$$

Substituting Eq. 48 back into Eq. 45, we have the closed-form solution of our objective:

$$(\mathbf{\Delta P})_{\text{SPEED}} = \mathbf{W}(\mathbf{C}_*\mathbf{C}_1^\top - \mathbf{C}_1\mathbf{C}_1^\top)\mathbf{P}\mathbf{Q}\mathbf{M}, \tag{49}$$

where $\mathbf{Q} = \mathbf{I} - \mathbf{M}\mathbf{C}_2(\mathbf{C}_2^\top \mathbf{P}\mathbf{M}\mathbf{C}_2)^{-1}\mathbf{C}_2^\top \mathbf{P}$ and $\mathbf{M} = (\mathbf{C}_1\mathbf{C}_1^\top \mathbf{P} + \mathbf{I})^{-1}$.

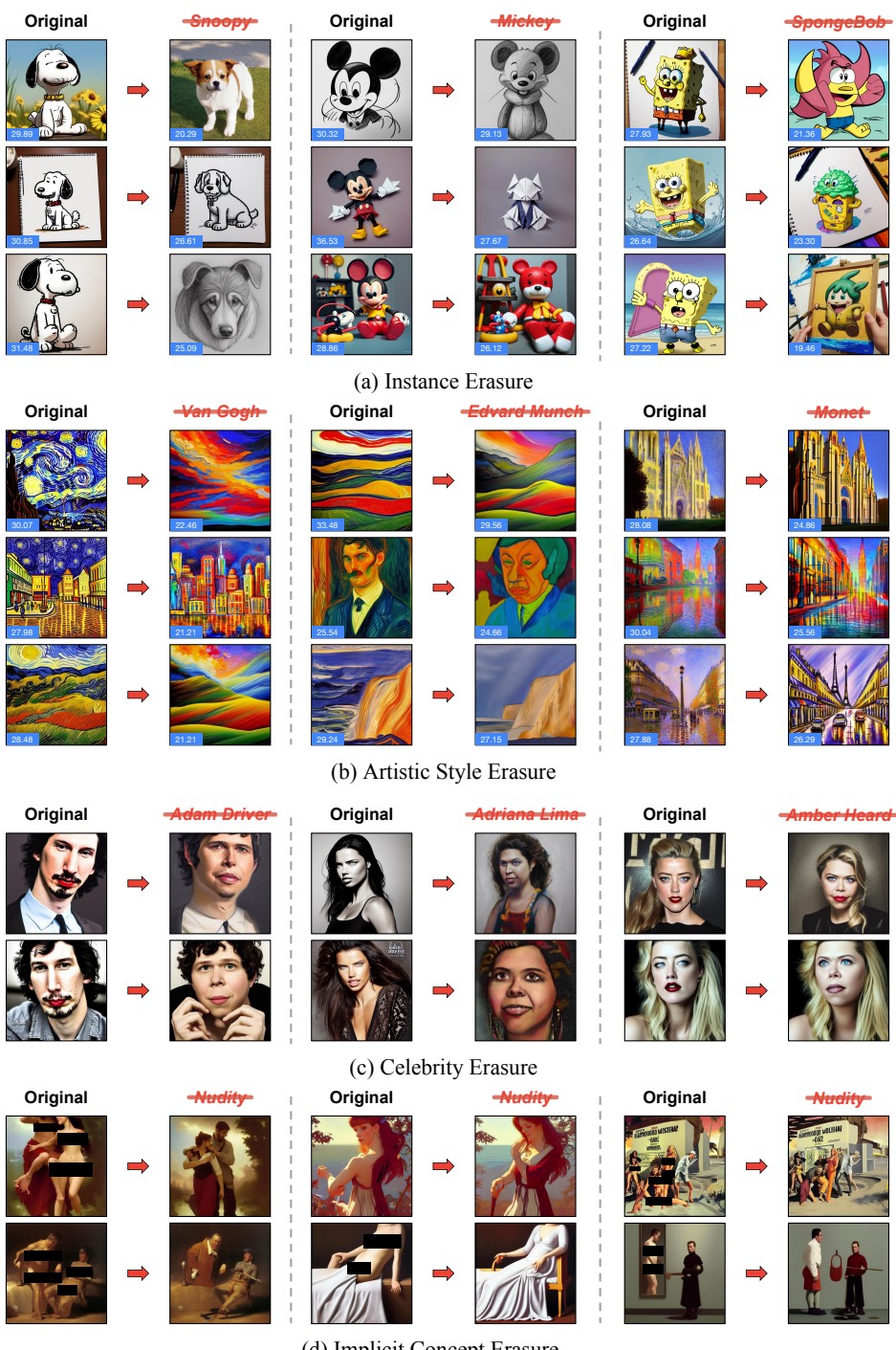

(a) Instance Erasure

(b) Artistic Style Erasure

(c) Celebrity Erasure

(d) Implicit Concept Erasure

Figure 7: **Qualitative demonstration of our erasure performance** across (a) *instance erasure,* (b) *artistic style erasure,* (c) *celebrity erasure,* and (d) *implicit concept erasure.* Our method achieves effective erasure efficacy across various scenarios while exhibiting superior prior preservation. The corresponding CS is highlighted in blue, indicating that successful erasure can be achieved without pushing CS much lower, as our results demonstrate sufficient erasure at a moderate level.

## C  IMPLEMENTATION DETAILS

### C.1  EXPERIMENTAL SETUP DETAILS

**Few-concept erasure.** We first compare methods on few-concept erasure, a fundamental concept erasure task, including both instance erasure and artistic style erasure following (Lyu et al., 2024). For instance erasure, we prepare 80 instance templates proposed in CLIP (Radford et al., 2021), such as *"a photo of the {Instance}"*, *"a drawing of the {Instance}"*, and *"a painting of the {Instance}"*. For artistic style erasure, we use ChatGPT (OpenAI, 2022; Achiam et al., 2023) to generate 30 artistic style templates, including *"{Artistic} style painting of the night sky with bold strokes"*, *"{Artistic} style landscape of rolling hills with dramatic brushwork"*, and *"Sunrise scene in {Artistic} style, capturing the beauty of dawn"*. Following (Lyu et al., 2024), we handpick the representative target and anchor concepts as the erasure set (*i.e.*, *Snoopy*, *Mickey*, *Spongebob* → ' ' in instance erasure and *Van Gogh*, *Picasso*, *Monet* → 'art' in artistic style erasure) and non-target concepts for evaluation (*i.e.*, *Pikachu* and *Hello Kitty* in instance erasure and *Paul Gauguin* and *Caravaggio* in artistic style erasure). In terms of the retain set, for instance erasure, we use a scraping script to crawl Wikipedia category pages to extract fictional character names and their page view counts with a threshold of 500,000 views from 2020.01.01 to 2023.12.31, resulting in 1,352 instances. For artistic style erasure, we use the 1,734 artistic styles collected from UCE (Gandikota et al., 2024). In evaluation, we generate 10 images per template per concept, resulting in 800 and 300 images for each concept in instance erasure and artistic style erasure, respectively. Moreover, we introduce the MS-COCO captions (Lin et al., 2014) to serve as general prior knowledge. In implementation, we use the first 1,000 captions to generate a total of 1000 images to compare CS and FID before and after erasure.

**Multi-concept erasure.** We then compare methods on multi-concept erasure, a more challenging and realistic concept erasure task. Following the experiment setup from (Lu et al., 2024a), we introduce a dataset consisting of 200 celebrities, where their portraits generated by SDv1.4 (Stability AI, 2022) can be recognizable with exceptional accuracy by the GIPHY Celebrity Detector (GCD) (Hasty et al., 2019). This dataset is divided into two groups: an erasure set with 10, 50, and 100 celebrities and a retain set with 100 other celebrities. The full list for both sets is presented in Table 4. We experiment with erasing 10, 50, and 100 celebrities with the predefined target concepts and the entire retain set is utilized in all cases. In evaluation, we prepare five celebrity templates, (*i.e.*, *"a portrait of {Celebrity}"*, *"a sketch of {Celebrity}"*, *"an oil painting of {Celebrity}"*, *"{Celebrity} in an official photo"*, and *"an image capturing {Celebrity} at a public event"*) and generate 500 images for both sets. For non-target concepts, we generate 1 image per template for each of the 100 concepts, totaling 500 images. For target concepts, we adjust the per-concept quantity to maintain a total of 500 images (*e.g.*, erasing 10 celebrities involves generating 10 images with 5 templates).

### C.2  ERASURE CONFIGURATIONS

**Implementation of previous works.** In our series of three concept erasure tasks, we mainly compare against four methods: ConAbl[4] (Kumari et al., 2023), MACE[5] (Lu et al., 2024a), RECE[6] (Gong et al., 2025), and UCE[7] (Gandikota et al., 2024), as they achieve SOTA performance across different concept erasure tasks. All the compared methods are implemented using their default configurations from the corresponding official repositories. One exception is that for MACE when erasing 50 celebrities, since it doesn't provide an official configuration and the *preserve weight* varies with the number of target celebrities, we set it to $1.2 \times 10^5$ to ensure a consistent balance between concept erasure and prior preservation.

**Implementation of SPEED.** In line with previous methods (Kumari et al., 2023; Lu et al., 2024a; Wu et al., 2025b; Gong et al., 2025; Gandikota et al., 2024), we edit the cross-attention (CA) layers within the diffusion model due to their role in text-image alignment (Hertz et al., 2022). In contrast, we only edit the value matrices in the CA layers, as suggested by (Wang et al., 2024b). This choice is grounded in the observation that the keys in CA layers typically govern the layout and compositional

---

[4] https://github.com/nupurkmr9/concept-ablation
[5] https://github.com/Shilin-LU/MACE
[6] https://github.com/CharlesGong12/RECE
[7] https://github.com/rohitgandikota/unified-concept-editing

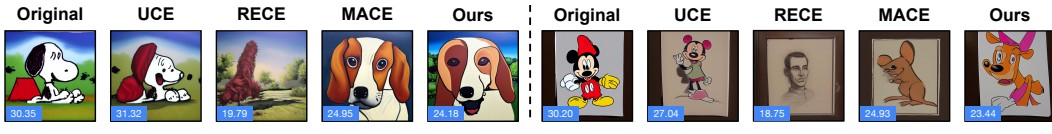

(a) ~~Snoopy~~                                    (b) ~~Mickey~~

Figure 8: **Comparison of CS values across different erasure methods.** We compare the results in erasing *Snoopy* and *Mickey*, and highlight the corresponding CS in blue. Our method achieves successful concept erasure with moderate CS values. In contrast, RECE achieves the lowest CS by enabling more aggressive erasure. For example, removing *Snoopy* into a landscape without a subject, and changing *Mickey* into a generic person. We argue that such over-erasure unnecessarily compromises prior preservation as evidenced by Tables 1 and 2.

structure of the attention map, while the values control the content and visual appearance of the images (Tewel et al., 2023). In the context of concept erasure, our goal is to effectively remove the semantics of the target concept, and we find that only editing the value matrices is sufficient in Figs. 4 and 5, which is quantitatively ablated in Appx. D.5. The augmentation times $N_A$ in Eq. 9 is set to 10 and the augmentation ranks $r$ in Eq. 7 is set to 1 as ablated in Appx. D.5. Meanwhile, given that eigenvalues are rarely strictly zero in practical applications when determining the null space, we select the singular vectors corresponding to the singular values below $10^{-1}$ on few-concept and implicit concept erasure and $10^{-4}$ on multi-concept erasure following (Fang et al., 2024).

# D ADDITIONAL EXPERIMENTS

## D.1 MORE DEMONSTRATIONS

We further provide qualitative visualizations of the erasure results in Fig. 7, illustrating the effectiveness of our method in performing precise and targeted concept erasure across diverse scenarios. Specifically, we showcase: (a) *instance erasure* from Table 1 (*left*); (b) *artistic style erasure* from Table 1 (*right*); (c) *celebrity erasure* from Table 2; and (d) *implicit concept erasure* (*e.g.*, *nudity*) from Table 9. In all cases, our method successfully removes the intended concept while preserving unrelated content, demonstrating its universal erasure applications.

We also evaluate the CS value before and after concept erasure to assess the erasure efficacy. As shown in Fig. 8, our method achieves successful erasure of specific concepts such as *Snoopy* and *Mickey* while maintaining moderate CS values (24.18 and 23.44, respectively). This indicates that effective erasure does not re-

Table 5: **Human study** of erasure efficacy in erasing *Snoopy*, *Mickey*, and *Spongebob* from Table 1.

|      | *Snoopy* | *Mickey* | *Spongebob* | Average |
| --- | --- | --- | --- | --- |
| RECE | 98.93% | 98.27% | 99.07% | 98.76% |
| Ours | 98.20% | 98.40% | 98.80% | 98.47% |

quire minimizing CS to an extreme. In contrast, RECE obtains the lowest CS (19.79 and 18.75), but this is achieved at the cost of overly aggressive erasure. For example, transforming *Snoopy* into an unrecognizable image and replacing *Mickey* with a generic human figure. While such strategies may enhance erasure efficacy, they also risk compromising prior knowledge. This trade-off is reflected in higher non-target FIDs, as shown in Tables 1 and 2.

To further demonstrate that our current erasure is adequate, we additionally conduct a human study. For our method and RECE, we randomly sample 50 generated images per method to erase *Snoopy*, *Mickey*, and *Spongebob*. We then recruit 30 human participants through Amazon Mechanical Turk (Amazon) to vote "*yes*" or "*no*" on whether the target concept is visually erased or not. As shown in Table 5, the overall results (RECE's 98.76% *v.s.* Our 98.47%) indicate that our method achieves successful erasure on par with RECE from the human perspective.

## D.2 COMPLETE RESULTS ON FEW-CONCEPT ERASURE

We present complete quantitative comparisons of few-concept erasure, including both CS and FID, in Table 6 and Table 7. Our results demonstrate that our method consistently achieves superior prior preservation, as indicated by higher CS and lower FID across the majority of non-target concepts.

Table 6: **Complete quantitative comparison of the few-concept erasure** in erasing instances from Table 1 (*left*). The best results are highlighted in **bold**, and grey columns are indirect indicators for measuring erasure efficacy on target concepts or prior preservation on non-target concepts.

| | *Snoopy* | | *Mickey* | | *Spongebob* | | *Pikachu* | | *Hello Kitty* | | MS-COCO | |
|---|---|---|---|---|---|---|---|---|---|---|---|---|
| | CS | FID | CS | FID | CS | FID | CS | FID | CS | FID | CS | FID |
| SD v1.4 | 28.51 | - | 26.62 | - | 27.30 | - | 27.44 | - | 27.77 | - | 26.53 | - |
| *Erase Snoopy* | | | | | | | | | | | | |
| | CS ↓ | FID ↑ | CS ↑ | FID ↓ | CS ↑ | FID ↓ | CS ↑ | FID ↓ | CS ↑ | FID ↓ | CS ↑ | FID ↓ |
| ConAbl | 25.44 | 98.38 | 26.63 | 37.08 | 26.95 | 38.92 | 27.47 | 26.14 | 27.65 | 36.52 | 26.40 | 21.20 |
| MACE | 20.90 | **165.74** | 23.46 | 105.97 | 23.35 | 102.77 | 26.05 | 65.71 | 26.05 | 75.42 | 26.09 | 42.62 |
| RECE | **18.38** | 151.46 | 26.62 | 26.63 | 27.23 | 34.42 | 27.47 | 21.99 | 27.78 | 32.35 | 26.39 | 25.61 |
| UCE | 23.19 | 102.86 | 26.64 | 24.87 | 27.29 | 29.86 | 27.47 | 19.06 | 27.75 | 27.86 | 26.46 | 22.18 |
| SPM | 23.72 | 116.26 | 26.62 | 31.21 | 27.21 | 31.96 | 27.41 | 19.82 | 27.80 | 30.95 | 26.47 | 20.71 |
| SPM w/o FT | 23.72 | 116.26 | 26.55 | 43.03 | 26.84 | 42.96 | 27.38 | 25.95 | 27.71 | 42.53 | 26.48 | 20.86 |
| Ours | 23.50 | 108.51 | 26.67 | **23.41** | 27.31 | **24.64** | 27.48 | **16.81** | 27.82 | **21.74** | 26.48 | **19.95** |
| *Erase Snoopy and Mickey* | | | | | | | | | | | | |
| | CS ↓ | FID ↑ | CS ↓ | FID ↑ | CS ↑ | FID ↓ | CS ↑ | FID ↓ | CS ↑ | FID ↓ | CS ↑ | FID ↓ |
| ConAbl | 25.26 | 106.78 | 26.58 | 57.05 | 26.81 | 45.08 | 27.34 | 35.57 | 27.74 | 41.48 | 26.42 | 24.34 |
| MACE | 20.53 | **170.01** | 20.63 | 142.98 | 22.03 | 112.01 | 24.98 | 91.72 | 23.64 | 106.88 | 25.50 | 55.15 |
| RECE | **18.57** | 150.84 | **19.14** | **145.59** | 27.29 | 35.85 | 27.37 | 26.05 | 27.71 | 40.77 | 26.31 | 30.30 |
| UCE | 23.60 | 99.30 | 24.79 | 86.32 | 27.32 | 30.58 | 27.38 | 23.51 | 27.74 | 31.76 | 26.38 | 26.06 |
| SPM | 23.18 | 122.17 | 22.71 | 117.30 | 26.92 | 38.35 | 27.35 | 27.13 | 27.76 | 39.61 | 26.45 | 24.59 |
| SPM w/o FT | 22.45 | 127.95 | 21.77 | 127.57 | 25.96 | 61.52 | 27.39 | 42.63 | 27.14 | 68.75 | 26.43 | 23.82 |
| Ours | 23.58 | 103.62 | 23.62 | 83.70 | 27.34 | 29.67 | 27.39 | **22.51** | 27.78 | **28.23** | 26.47 | **23.66** |
| *Erase Snoopy and Mickey and Spongebob* | | | | | | | | | | | | |
| | CS ↓ | FID ↑ | CS ↓ | FID ↑ | CS ↓ | FID ↑ | CS ↑ | FID ↓ | CS ↑ | FID ↓ | CS ↑ | FID ↓ |
| ConAbl | 24.92 | 112.66 | 26.46 | 63.95 | 25.12 | 102.68 | 27.36 | 46.47 | 27.72 | 48.24 | 26.37 | 26.71 |
| MACE | 19.86 | **175.43** | 19.35 | 140.13 | 20.12 | 143.17 | 19.76 | 110.12 | 21.03 | 128.56 | 23.39 | 66.39 |
| RECE | **18.17** | 155.26 | **18.87** | **149.77** | **16.23** | **178.55** | 27.34 | 40.52 | 27.71 | 52.06 | 26.32 | 32.51 |
| UCE | 23.29 | 101.40 | 24.63 | 88.11 | 19.08 | 140.40 | 27.45 | 29.20 | 27.82 | 38.15 | 26.30 | 28.71 |
| SPM | 22.86 | 125.66 | 22.08 | 123.20 | 20.92 | 153.36 | 27.45 | 37.51 | 27.63 | 46.63 | 26.48 | 25.47 |
| SPM w/o FT | 21.80 | 137.98 | 20.86 | 139.48 | 20.19 | 163.21 | 26.68 | 66.15 | 26.24 | 85.35 | 26.33 | 25.05 |
| Ours | 23.69 | 103.33 | 23.93 | 86.55 | 21.39 | 109.28 | 27.47 | **21.40** | 27.76 | **26.22** | 26.51 | **24.99** |

Figure 9: **Qualitative comparison with SPM and SPM w/o FT** in erasing single and multiple instances. The erased and preserved generations are highlighted with red and green boxes, respectively. Our method demonstrates superior prior preservation compared to both baselines. Meanwhile, without *Facilitated Transport*, SPM w/o FT shows poorer prior preservation in multi-concept erasure (*e.g.*, marked by ⊗) with significant semantic changes compared to original generations.

## D.3 COMPARISON ON MORE BASELINES

In this section, we compare against more methods because of the page limit in our main paper, including ESD[8] (Gandikota et al., 2023), RACE[9] (Kim et al., 2024), Receler[10] (Huang et al., 2024),

---

[8] https://github.com/rohitgandikota/erasing
[9] https://github.com/chkimmmmm/R.A.C.E.
[10] https://github.com/jasper0314-huang/Receler

Table 7: **Complete quantitative comparison of the few-concept erasure** in erasing artistic styles from Table 1 (*right*). The best results are highlighted in **bold**, and grey columns are indirect indicators for measuring erasure efficacy on target concepts or prior preservation on non-target concepts.

| | Van Gogh | | Picasso | | Monet | | Paul Gauguin | | Caravaggio | | MS-COCO | |
|---|---|---|---|---|---|---|---|---|---|---|---|---|
| | CS | FID | CS | FID | CS | FID | CS | FID | CS | FID | CS | FID |
| SD v1.4 | 28.75 | - | 27.98 | - | 28.91 | - | 29.80 | - | 26.27 | - | 26.53 | - |
| *Erase Van Gogh* | | | | | | | | | | | | |
| | CS ↓ | FID ↑ | CS ↑ | FID ↓ | CS ↑ | FID ↓ | CS ↑ | FID ↓ | CS ↑ | FID ↓ | CS ↑ | FID ↓ |
| ConAbl | 28.16 | 129.57 | 27.07 | 77.01 | 28.44 | 63.80 | 29.49 | 63.20 | 26.15 | 79.25 | 26.46 | **18.36** |
| MACE | 26.66 | 169.60 | 27.39 | 69.92 | 28.84 | 60.88 | 29.39 | 56.18 | 26.19 | 69.04 | 26.50 | 23.15 |
| RECE | 26.39 | 171.70 | 27.58 | 60.57 | 28.83 | 61.09 | 29.58 | 47.07 | 26.21 | 72.85 | 26.52 | 23.54 |
| UCE | 28.10 | 133.87 | 27.70 | 43.02 | 28.92 | 40.49 | 29.62 | 32.62 | 26.23 | 61.72 | 26.54 | 19.63 |
| ESD-X | 27.04 | 200.05 | 26.50 | 111.07 | 28.14 | 90.35 | 29.45 | 106.70 | 25.70 | 107.85 | 26.10 | 33.19 |
| ESD-U | 26.24 | 205.06 | 26.28 | 153.10 | 27.79 | 105.78 | 29.59 | 164.83 | 26.14 | 124.41 | 26.35 | 38.08 |
| RACE | 23.03 | 233.25 | 25.54 | 127.28 | 26.44 | 94.49 | 27.78 | 106.43 | 25.08 | 114.94 | 25.92 | 41.52 |
| Receler | **21.53** | 245.40 | 24.88 | 134.35 | 23.61 | 143.17 | 25.02 | 194.58 | 24.52 | 133.94 | 25.95 | 37.00 |
| Ours | 26.29 | 131.02 | 27.96 | **35.86** | 28.94 | **16.85** | 29.71 | **24.94** | 26.24 | **39.75** | **26.55** | 20.36 |
| *Erase Picasso* | | | | | | | | | | | | |
| | CS ↑ | FID ↓ | CS ↓ | FID ↑ | CS ↑ | FID ↓ | CS ↑ | FID ↓ | CS ↑ | FID ↓ | CS ↑ | FID ↓ |
| ConAbl | 28.66 | 60.44 | 26.97 | 131.45 | 28.72 | 36.23 | 29.68 | 65.23 | 26.20 | 79.12 | 26.43 | 20.02 |
| MACE | 28.68 | 59.58 | 26.48 | 137.09 | 28.73 | 37.02 | 29.71 | 46.35 | 26.23 | 66.20 | 26.47 | 22.86 |
| RECE | 28.71 | 51.09 | 26.66 | 126.40 | 28.87 | 25.39 | 29.69 | 46.08 | 26.22 | 75.61 | 26.48 | 23.03 |
| UCE | 28.72 | 37.58 | 26.99 | 102.21 | 28.92 | **16.72** | 29.71 | 32.48 | 26.22 | 59.27 | 26.50 | 20.33 |
| ESD-X | 28.58 | 104.48 | 26.07 | 178.18 | 28.32 | 62.79 | 29.31 | 96.70 | 25.84 | 100.54 | 26.15 | 34.12 |
| ESD-U | 28.69 | 109.39 | 26.47 | 156.35 | 28.64 | 67.69 | 29.64 | 95.39 | 26.04 | 105.76 | 26.35 | 35.78 |
| RACE | 28.12 | 112.29 | 24.84 | 185.78 | 27.88 | 72.79 | 28.91 | 93.19 | 25.81 | 110.23 | 25.77 | 42.01 |
| Receler | 25.92 | 199.56 | **23.10** | 243.28 | 26.92 | 94.89 | 26.51 | 208.01 | 25.34 | 135.35 | 25.88 | 37.20 |
| Ours | 28.76 | 19.18 | 26.22 | 117.71 | 28.88 | 19.87 | 29.75 | **24.73** | 26.24 | 43.63 | 26.51 | **19.98** |
| *Erase Monet* | | | | | | | | | | | | |
| | CS ↑ | FID ↓ | CS ↑ | FID ↓ | CS ↓ | FID ↑ | CS ↑ | FID ↓ | CS ↑ | FID ↓ | CS ↑ | FID ↓ |
| ConAbl | 28.58 | 68.77 | 27.43 | 64.25 | 27.05 | 96.67 | 29.09 | 57.33 | 26.09 | 71.88 | 26.45 | 21.03 |
| MACE | 28.56 | 61.50 | 27.74 | 48.41 | 25.98 | 116.34 | 29.39 | 49.66 | 25.98 | 65.87 | 26.47 | 22.76 |
| RECE | 28.63 | 56.26 | 27.88 | 45.97 | 25.87 | 121.28 | 29.63 | 46.38 | 26.16 | 64.19 | 26.49 | 24.94 |
| UCE | 28.65 | 42.25 | 27.91 | **38.73** | 27.12 | 98.37 | 29.58 | 33.00 | 26.16 | 56.49 | **26.51** | 21.58 |
| ESD-X | 28.15 | 115.51 | 26.56 | 92.69 | 25.97 | 124.90 | 28.85 | 89.07 | 25.92 | 102.53 | 25.98 | 35.79 |
| ESD-U | 28.73 | 134.10 | 26.87 | 114.64 | 25.15 | 134.02 | 29.44 | 135.64 | 25.72 | 131.90 | 26.21 | 38.16 |
| RACE | 27.13 | 132.42 | 25.99 | 106.70 | 23.08 | 149.16 | 27.52 | 98.71 | 24.96 | 110.34 | 25.81 | 41.96 |
| Receler | 24.94 | 169.55 | 26.16 | 105.24 | **21.06** | 182.34 | 24.81 | 199.23 | 25.03 | 122.42 | 25.99 | 36.39 |
| Ours | 28.76 | **28.78** | 27.93 | 41.21 | 25.06 | 134.11 | 29.66 | **27.85** | 26.22 | 55.20 | 26.48 | **20.87** |

Table 8: **Quantitative comparison with SPM and SPM w/o FT in multi-concept erasure.** The best results are highlighted in **bold**. Our method is capable of erasing up to 100 celebrities at once with low $\text{Acc}_e$ (%) and preserving other non-target celebrities with less appearance alteration with high $\text{Acc}_r$ (%), resulting in the best overall erasure performance $H_o$ (shaded in pink). FAIL indicates that the model collapses with noisy generations ($\text{Acc}_e = \text{Acc}_r = 0.00\%$).

| | Erase 10 Celebrities | | | MS-COCO | | Erase 50 Celebrities | | | MS-COCO | | Erase 100 Celebrities | | | MS-COCO | |
|---|---|---|---|---|---|---|---|---|---|---|---|---|---|---|---|
| | $\text{Acc}_e$ ↓ | $\text{Acc}_r$ ↑ | $H_o$ ↑ | CS ↑ | FID ↓ | $\text{Acc}_e$ ↓ | $\text{Acc}_r$ ↑ | $H_o$ ↑ | CS ↑ | FID ↓ | $\text{Acc}_e$ ↓ | $\text{Acc}_r$ ↑ | $H_o$ ↑ | CS ↑ | FID ↓ |
| SD v1.4 | 91.99 | 89.66 | 14.70 | 26.53 | - | 93.08 | 89.66 | 12.85 | 26.53 | - | 90.18 | 89.66 | 17.70 | 26.53 | - |
| SPM | 0.00 | 51.79 | 68.24 | 26.42 | 48.44 | 0.00 | 0.00 | FAIL | 26.32 | 52.61 | 0.00 | 0.00 | FAIL | 25.15 | 63.20 |
| SPM w/o FT | 0.00 | 5.08 | 9.68 | 26.38 | 52.23 | 0.00 | 0.00 | FAIL | 16.22 | 170.68 | 0.00 | 0.00 | FAIL | 14.34 | 245.92 |
| Ours | 1.81 | 89.09 | **93.42** | 26.47 | 30.02 | 3.46 | 88.48 | **92.34** | 26.46 | 39.23 | 5.87 | 85.54 | **89.63** | 26.22 | 44.97 |

and SPM[11] (Lyu et al., 2024). While the first three methods are training-based, focusing solely on modifying model parameters, SPM not only fine-tunes the model weights using LoRA (Hu et al., 2021) but also intervenes in the image generation process through *Facilitated Transport*. Specifically, this module dynamically adjusts the LoRA scale based on the similarity between the sampling prompt and the target concept. In other words, if the prompt contains the target concept or is highly

---

[11] https://github.com/Con6924/SPM

Table 9: **Evaluation of implicit concept erasure** in erasing *nudity* on four benchmarks. We report the Attack Success Rate (ASR) detected by NudeNet with a threshold of 0.6. ✓ and × indicate whether the method can defend against white-box attacks, respectively. The best and second-best results are marked in **bold** and underlined.

| | I2P | MMA | Ring-A-Bell | UnlearnDiff | Time (s) ↓ | MS-COCO CS | MS-COCO FID | White-Box Attack |
|---|---|---|---|---|---|---|---|---|
| MACE (Lu et al., 2024a) | 0.21 | 0.04 | 0.05 | 0.67 | 55 (×15) | 24.06 | 52.78 | ✓ |
| CPE (Lee et al., 2025b) | 0.07 | 0.01 | **0.00** | - | 500 (×138) | 26.32 | 48.23 | × |
| AdvUnlearn (Zhang et al., 2024b) | **0.04** | **0.00** | **0.00** | **0.21** | 15860 (×4400) | 24.05 | 57.22 | ✓ |
| UCE (Gandikota et al., 2024) | 0.24 | 0.38 | 0.39 | 0.80 | 1.2 (×0.33) | 26.24 | 38.60 | ✓ |
| RECE (Gong et al., 2025) | 0.14 | 0.20 | 0.18 | 0.65 | 1.5 (×0.41) | 25.98 | 40.37 | ✓ |
| RACE (Kim et al., 2024) | 0.23 | 0.29 | 0.21 | 0.47 | 2910 (×800) | 25.54 | 42.73 | ✓ |
| Receler (Huang et al., 2024) | 0.13 | 0.07 | 0.01 | - | 5560 (×1500) | 25.93 | 40.29 | × |
| Ours w/o AT | 0.20 | 0.24 | 0.20 | 0.75 | 3.6 (×1) | 26.29 | 37.82 | ✓ |
| Ours w/ AT | 0.10 | 0.01 | **0.00** | 0.45 | 4.5 (×1.25) | 26.03 | 39.51 | ✓ |

Table 10: **Ablation study on the edited parameters.** Our scheme on only editing the value matrices achieves a superior balance between erasure efficacy (*e.g.*, target CS of 26.29) and prior preservation (*e.g.*, the lowest FIDs across all non-target concepts).

| Ablation | Parameters Key | Parameters Value | *Van Gogh* CS ↓ | *Picasso* FID ↓ | *Monet* FID ↓ | *Paul Gauguin* FID ↓ | *Caravaggio* FID ↓ | MS-COCO CS ↑ | MS-COCO FID ↓ |
|---|---|---|---|---|---|---|---|---|---|
| 1 | ✓ | × | 27.67 | 42.11 | 26.09 | 28.08 | 52.44 | **26.55** | **18.72** |
| 2 | ✓ | ✓ | **26.24** | 48.41 | 28.65 | 33.79 | 57.23 | 26.53 | 23.20 |
| Ours | × | ✓ | 26.29 | **35.86** | **16.85** | **24.94** | **39.75** | **26.55** | 20.36 |

relevant, this scale is set to a large value, whereas if there is little to no relevance, it is set close to 0, functioning similarly to a text filter. We argue that such a comparison with SPM is not fair since we only focus on modifying the model parameters, and therefore, we compare both the original SPM and SPM without *Facilitated Transport* (SPM w/o FT) for a fair comparison. In the latter version, the LoRA scale is set to 1 by default.

The quantitative comparative results are shown in Tables 6 and 7, where our method consistently achieves the best prior preservation compared to all compared baselines. Even equipped with *Facilitated Transport* (*i.e.*, SPM w/ FT), our method achieves the lowest non-target FID (*e.g.*, on *Pikachu* and *Hello Kitty*). This superiority amplifies as the number of target concepts increases as shown in Table 8. For example, with the number of target concepts increasing from 1 to 3, our FID in *Pikachu* rises from 16.81 to 21.40 (4.59 ↑), while SPM increases from 19.82 to 37.51 (17.69 ↑), where a similar pattern is observed in *Hello Kitty* (Our 4.48 ↑ *v.s.* SPM's 15.68 ↑). Once removing the *Facilitated Transport* module, SPM w/o FT shows poorer prior preservation with rapidly increasing FIDs (highlighted in red in Table 6). More qualitative results are shown in Fig. 9.

### D.4 ON IMPLICIT CONCEPT ERASURE

**Evaluation setup.** We evaluate the erasure efficacy on implicit concepts (*e.g.*, *nudity*), where the target concept does not explicitly appear in the text prompt. We conduct experiments on the In-appropriate Image Prompt (I2P) benchmark (Schramowski et al., 2023), which consists of various implicit inappropriate prompts involving violence, sexual content, and nudity. To evaluate adversarial robustness, we further introduce three adversarial attack benchmarks, including two black-box benchmarks (MMA (Yang et al., 2024b) and Ring-A-Bell (Tsai et al., 2023)) and one white-box benchmark (UnlearnDiff (Zhang et al., 2024c)). For concept erasure, we follow the setting in (Gong et al., 2025) to erase *nudity* → ' '. During evaluation, we use NudeNet (Bedapudi, 2019) with a threshold of 0.6 to detect nude content and report the Attack Success Rate (ASR). Moreover, since the retain set only includes ' ' in implicit concept erasure, we add an identity matrix $\mathbf{I}$ with weight $\lambda = 0.5$ to the term $(\mathbf{C}_2^\top \mathbf{PMC}_2)^{-1}$ in Eq. 12 to ensure invertibility.

**Analysis and discussion.** In addition to the aforementioned methods, we introduce more adversarial training-based methods for comparison, including CPE (Lee et al., 2025b), AdvUnlearn (Zhang et al., 2024b), RACE (Kim et al., 2024), and Receler (Huang et al., 2024). These methods enhance robustness against adversarial attacks by explicitly incorporating adversarial training objectives. We

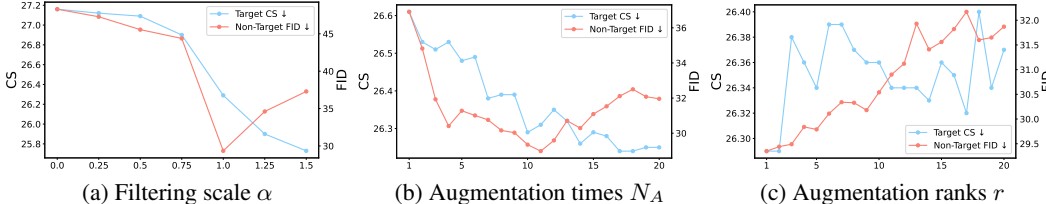

Figure 10: **Ablation study on hyperparameters.** We report target CS of erasing *Van Gogh* and non-target FID averaged over other four styles (*i.e.*, *Picasso, Monet, Paul Gauguin, Caravaggio*).

also adapt our method with adversarial training/editing (denoted as Ours w/ AT) following the setting in RECE (Gong et al., 2025) to provide a fair comparison. As shown in Table 9, we observe that adversarial training-based methods such as CPE and AdvUnlearn achieve strong erasure efficacy but incur extremely high computational costs. Editing-based approaches like UCE and RECE are more efficient yet less robust under both black-box and white-box attacks. Notably, CPE and Receler rely on additional modules for concept erasure, which makes them particularly vulnerable in the white-box setting since attackers can directly exploit these components to bypass erasure. In contrast, our method without adversarial training (Ours w/o AT) already offers a favorable balance between efficiency and prior preservation, and extending it with adversarial training/editing (Ours w/ AT) further improves robustness, reducing ASR across all benchmarks and lowering the white-box UnlearnDiff score from 0.75 to 0.45 while maintaining competitive runtime and prior knowledge preservation.

### D.5 ABLATION STUDIES

**Edited parameters.** We compare the impact on editing different CA parameters in Table 10 and draw the following conclusions: (1) Only editing the key matrices cannot achieve effective erasure, with the target CS being 27.67 (*v.s.* the original CS of 28.75). This is because they mainly arrange the layout information of the generation and cannot effectively erase the semantics of the target concept. (2) Simultaneously editing both the key and value matrices can achieve effective erasure, but it will also excessively damage prior knowledge. (3) Only editing the value matrices achieves a superior balance between erasure efficacy and prior preservation. Compared to Ablation 2, the editing of key matrices leads to excessive erasure, which is unnecessary in concept erasure.

**Filtering scale.** We ablate the filtering threshold scale in the Influence-based Prior Filtering (IPF) module in Sec. 4.1 by scaling the impact scores $\mu$ with a factor $\alpha$, which controls the strength of filtering influential priors. As shown in Fig. 10 (a), varying $\alpha$ directly affects the trade-off between erasure efficacy and prior preservation. When $\alpha$ is small (*i.e.*, close to 0), more weakly affected priors are included in the retain set, increasing its rank and overly shrinking the null space. This leads to worse erasure efficacy (higher CS) and poor preservation (higher FID). Conversely, a higher $\alpha$ yields better erasure performance due to fewer retain concepts, but still increases the FID because of non-comprehensive prior coverage. The best balance is observed at moderate thresholds (*e.g.*, $\alpha = 1$ in our setup), achieving both effective erasure and competitive prior preservation.

**Augmentation times.** We ablate the augmentation times $N_A$ proposed in the Directed Prior Augmentation (DPA) module in Sec. 4.2, which controls the balance between semantic degradation and retain coverage along with the Influence-based Prior Filtering (IPF) module. It can be observed from Fig. 10 (b) that: (1) As $N_A$ increases, the non-target FID exhibits a trend of first decreasing and then increasing. This suggests that when $N_A$ is small (*i.e.*, $1 \rightarrow 10$), augmenting existing non-target concepts with semantically similar concepts facilitates a more comprehensive retain coverage, thereby improving prior preservation. However, when $N_A$ exceeds a certain threshold (*i.e.*, $10 \rightarrow 20$), further augmentation of non-target concepts leads to narrowing the null-space derivation with semantic degradation, ultimately degrading prior preservation. (2) Target CS generally shows a declining trend, indicating that the proposed Prior Knowledge Refinement strategy not only improves prior preservation but also exerts a positive impact on erasure efficacy.

**Augmentation ranks.** Another hyperparameter to be ablated is the augmentation ranks $r$. From Eq. 7, we introduce the number of the smallest singular values, *i.e.*, augmentation ranks $r$ in deriving $\mathbf{P}_{\min} = \mathbf{U}_{\min}\mathbf{U}_{\min}^{\top}$ with $\mathbf{U}_{\min} = \mathbf{U}_{\mathbf{W}}[:, -r :]$. Mathematically, $r$ represents the directions in which the DPA module can augment in the concept embedding space and constrains the rank of the

augmented embeddings to a maximum of $r$. As shown in Fig. 10 (c), as $r$ increases, the non-target FID exhibits an overall upward trend, indicating that introducing more ranks does not benefit prior preservation, as it narrows the null space. At the same time, as shown in Table 3, such augmentation by DPA also remains necessary, as it enables more comprehensive coverage of non-target knowledge with semantically similar concepts, leading to improved prior preservation.

# E   LLM USAGE STATEMENT

We use large language models (LLMs) as an auxiliary tool during preparing this work. The LLM is employed to generate artistic style templates for the artistic style erasure task (see Appx. C.1) and to refine the clarity and readability of certain parts of the manuscript, such as polishing grammar, improving fluency, and standardizing terminology. In addition, LLMs are occasionally used to suggest alternative phrasings when writing sections like the introduction and related work, but the final narrative, arguments, and presentation choices are made solely by the authors. All methodological ideas, theoretical derivations, experiment designs, and analyses are developed independently by the authors without assistance from LLMs. We do not rely on LLMs for generating novel research ideas, conducting experiments, interpreting results, or writing technical content. The role of LLMs is purely supportive and limited to stylistic refinement and auxiliary text generation, and thus they are not regarded as scientific contributors to this paper.

# F   LIMITATION

Despite the promising results, SPEED is designed with linear null-space projections, which may not fully capture the nonlinear interactions between concepts in large diffusion models. In practice, this can lead to imperfect preservation when erasing highly entangled or stylistically subtle concepts. In addition, our evaluation mainly covers benchmarks with explicit or implicit concepts; the effectiveness on more abstract (*e.g.*, *freedom*), compositional (*e.g.*, *a blue cat*), or cultural (*e.g.*, *Día de los Muertos*) concepts remains less explored. Finally, although our method scales efficiently to 100 concepts, extending it to even larger-scale or continual erasure may require additional mechanisms.

