# OpenReview forum: "SPEED: Scalable, Precise, and Efficient Concept Erasure for Diffusion Models"
_ICLR.cc/2026/Conference — ICLR 2026 Poster_

### Official Review · Reviewer_NrN5 · 2025-10-20

**Soundness:** 3
**Presentation:** 3
**Contribution:** 3
**Rating:** 6
**Confidence:** 3

**Summary:**

This paper proposes a SPEED framework for erasing unwanted concepts in diffusion models. Specifically, this paper aims to develop a scalable, precise, and efficient concept erasure method. By formulating the concept erasure task into a null-space constrained optimization problem, the proposed framework could effectively remove the unwanted concepts while successfully preserving the unrelated concepts. This paper also proposes auxiliary strategies, including Influence-based Prior Filtering (IPF), Directed Prior Augmentation (DPA), and Invariant Equality Constraints (IEC), to prevent severe semantic degradation and improve retent set coverage. The experiments are conducted on various concept erasure tasks.

**Strengths:**

1. This paper aims to enhance multiple crucial aspects of concept erasing tasks, such as scaling the setting to handle large-scale multi-concept erasing, precisely removing the target concepts solely, and reducing computation time. The proposed method facilitates the development of trustworthy generative AI.
2. The proposed SPEED framework is reasonable and technically sound.
3. The overall paper is easy to follow and well organized.

**Weaknesses:**

1. While this paper demonstrates the ability to effectively erasing the target concepts while preserving the unrelated concepts, it remains unclear whether the model could handle the paraphrased prompts, which is called robustness issues. It would lead to the target concepts being easily recovered by the paraphrase prompts with the same or similar semantics.
2. In Figure 6, it appears that background changes in the output images. It seems removing the target concepts still affects the other parts of the images.

**Questions:**

1. In Figure 6(b), this approach shows the ability to perform model knowledge editing to edit the target concept to a specific one. Can this ability be extended to multi-concept knowledge editing?

---

> ### Author Response · Authors · 2025-11-21
> **Author Response**
>
> > W1: While this paper demonstrates the ability to effectively erasing the target concepts while preserving the unrelated concepts, it remains unclear whether the model could handle the paraphrased prompts, which is called robustness issues. It would lead to the target concepts being easily recovered by the paraphrase prompts with the same or similar semantics.
> >
>
> Thank you for raising this important point. We agree that robustness to paraphrased prompts is essential, as erased concepts may otherwise reappear through semantically equivalent variations. We have already conducted comprehensive robustness evaluations in **Appx. D.4**, covering “implicit concept erasure” (I2P) as well as “black-box“ (MMA, Ring-A-Bell) and ”white-box“ (UnlearnDiff) adversarial attacks. Here, ”implicit erasure” tests whether concept erasure holds when the prompt expresses the concept implicitly, while “black-box” and “white-box“ attacks generate paraphrased or adversarially rewritten prompts to revive the erased concept, effectively probing paraphrase-level robustness. As show in Table 8, without adversarial training (AT), SPEED w/o AT already outperforms other non-adversarial-training baselines (e.g., UCE), and after applying adversarial training, SPEED w/ AT achieves robustness comparable to state-of-the-art adversarial-training methods (e.g., CPE, AdvUnlearn, RECE, RACE, and Receler), while maintaining substantially lower runtime (4.5 seconds).
>
> ---
>
> > W2: In Figure 6, it appears that background changes in the output images. It seems removing the target concepts still affects the other parts of the images.
> >
>
> Thank you for your thoughtful comment. We would like to clarify that **SPEED is capable of preserving non-target semantics (i.e., other parts of the image) specified in the prompt**, and does not alter unrelated semantics during erasure. The background variation observed in Fig. 6 (a) is caused by the prompt itself. The prompt here (“*Snoopy in Van Gogh style*”) does not mention any background. As a result, the diffusion model generates a random background for each image. After erasure, the background may also change because it is not constrained by the prompt, rather than due to the limitation of SPEED in preserving non-target information.
>
> To address this concern, we include more detailed visualizations in the revised paper (Fig. 11) for both instance erasure and artistic style erasure. These results clearly demonstrate that SPEED preserves the non-target semantics (e.g., background information), whenever such content is present in the prompt. This further highlights the precision of our method in removing only the target concept while retaining all other content.
>
> ---
>
> > Q1: In Figure 6 (b), this approach shows the ability to perform model knowledge editing to edit the target concept to a specific one. Can this ability be extended to multi-concept knowledge editing?
> >
>
> Thank you for your question. Yes, our method can also extend to multi-concept knowledge editing. Since SPEED formulates concept erasure through a null-space constrained parameter update, it can simultaneously map multiple target concepts to user-specified anchors without additional architectural changes. In the revised paper (Fig. 12), we have included additional visual examples demonstrating multi-concept editing, and we will expand these results and explore more application scenarios in future work.

---

### Official Review · Reviewer_oMhk · 2025-10-30

**Soundness:** 3
**Presentation:** 3
**Contribution:** 3
**Rating:** 6
**Confidence:** 3

**Summary:**

This paper identifies a null space to enable multi-concept erasure while preserving non-targeted concepts. To minimize the impact on non-target concepts, it selects the most affected ones based on the prior shift, which quantifies how parameter updates perturb each non-target concept. Additionally, it augments the non-target concepts by perturbing them with directed noise, guiding their embeddings toward closer semantic representations. Furthermore, it imposes constraints on certain invariants during the T2I generation process. Experimental results show that the proposed method achieves superior performance in a short amount of time.

**Strengths:**

1. The null space–based method demonstrates a strong ability to preserve untargeted concepts, and the closed-form solution offers high computational efficiency.
2. The paper is well-organized. To enable accurate null-space construction, the authors use concept selection to reduce the rank of the correlation matrix. To address the resulting limited retain coverage, they further augment non-target embeddings using directed noise.

**Weaknesses:**

The main weakness lies in the CLIP Score, a common metric for evaluating concept erasure.
The results show that SPEED does not achieve the lowest CLIP Score. Although the authors provide visualizations demonstrating that erasure remains effective without reaching the minimum score, some residual patterns can still be observed. For example, in the left figure of Fig. 7(c), the mouth remains similar after removal, indicating instability in the concept erasure process.
Also, the concept erasure results in Table 2 show that SPEED is less effective at removing target concepts compared to other methods.

**Questions:**

Could adjusting the number of non-target concepts improve concept erasure performance and achieve a better trade-off between erasure and preservation of non-target concepts?

---

> ### Author Response · Authors · 2025-11-21
> **Author Response (Part 1/2)**
>
> > W1: The main weakness lies in the CLIP Score, a common metric for evaluating concept erasure. The results show that SPEED does not achieve the lowest CLIP Score. Although the authors provide visualizations demonstrating that erasure remains effective without reaching the minimum score, some residual patterns can still be observed. For example, in the left figure of Fig. 7(c), the mouth remains similar after removal, indicating instability in the concept erasure process. Also, the concept erasure results in Table 2 show that SPEED is less effective at removing target concepts compared to other methods.
> >
>
> We thank the reviewer for the thoughtful comments. Our response focuses on the following three key points:
>
> (1) While our method does not always achieve the lowest CLIP Score (CS) on target concepts in instance erasure, we illustrate that our erasure is already sufficient in Figs. 4-7. As further discussed in Appx. D.1 (Fig. 8), RECE reaches lower CS by suppressing the other semantics (e.g., removing *Snoopy* into a landscape without a subject, and changing *Mickey* into a generic person), but this comes at the cost of noticeable degradation in prior preservation (by comparing RECE and UCE in Table 1). To further demonstrate that our current erasure is adequate, we additionally conduct a human study. For our method and RECE, we randomly sample 50 generated images per method to erase *Snoopy and Mickey, and SpongeBob,* and ask human evaluators to determine whether the target concept’s semantics remain. We recruited 30 human participants through Amazon Mechanical Turk to vote “yes” or “no” on whether the concept is visually erased or not. The final erasure success rates (%) are reported in the table below. The overall results (RECE’s 98.76% v.s. Our 98.47%) indicate that our method achieves successful erasure on par with RECE from the human perspective. The human study details and final results are added to our revised paper in Appx. D.1.
>
> | **Erasure Success Rate (%)** | ***Erase ~~Snoopy~~ and ~~Mickey~~ and ~~Spongebob~~*** |  |  |  |
> | --- | --- | --- | --- | --- |
> |  | Snoopy | Mickey | Spongebob | Average |
> | **RECE** | 98.93% | 98.27% | 99.07% | 98.76% |
> | **Ours** | 98.20% | 98.40% | 98.80% | 98.47% |
>
> (2) Our method, SPEED, is designed to be **scalable**, **precise**, and **efficient**, which provides clear advantages over existing approaches.
>
> - Compared with RECE, our method is superior primarily in scalability and precision. RECE’s adversarial editing paradigm indeed achieves effective erasure, but this comes at the cost of degrading prior preservation. As the number of target concepts increases, the deterioration of prior preservation gradually accumulates. As shown in Table 2, RECE’s MS-COCO CS drops from 16.75 (erasing 10 concepts) to 13.49 (erasing 50 concepts) and further to 12.09 (erasing 100 concepts), which generates nearly meaningless noisy images for general concepts. In contrast, our method achieves MS-COCO CS from 26.47 (erasing 10 concepts) to 26.46 (erasing 50 concepts) to 26.22 (erasing 100 concepts), while still ensuring strong prior preservation after erasing 100 concepts.
> - Compared with MACE, the current SOTA multi-concept erasure method, our advantage lies in both efficiency and multi-concept performance. MACE requires about **30 minutes** to erase 100 concepts with a harmonic mean score of $H_o = 87.1$, while our closed-form solution enables direct computation of the model update and achieves erasure of 100 concepts in **only 5 seconds** with $H_o = 89.6$.
>
> (3) Considering strong erasure efficacy is necessary in certain scenarios (e.g., adversarial attacks), in Appx. D.4, we further introduce a variant of our method (i.e., SPEED w/ AT), which incorporates the adversarial training (AT) strategy from RECE into our method. As shown in Table 8, without adversarial training, SPEED w/o AT already outperforms other non–adversarial-training baselines (e.g., UCE); after applying adversarial training, SPEED w/ AT surpasses RECE and MACE in all benchmarks, and achieves comparable robustness to state-of-the-art methods such as CPE, AdvUnlearn, RACE, and Receler, maintaining substantially lower runtime at only 4.5 seconds.

---

> ### Author Response · Authors · 2025-11-21
> **Author Response (Part 2/2)**
>
> > Q1: Could adjusting the number of non-target concepts improve concept erasure performance and achieve a better trade-off between erasure and preservation of non-target concepts?
> >
>
> Thank you for the question. Intuitively, adjusting the number of non-target concepts does affect the erasure–preservation trade-off. Since our method is based on null-space construction, a very large retain set increases the rank of the correlation matrix and shrinks the null space, making preservation more difficult, while an overly small retain set fails to capture the broad prior knowledge that should be preserved.
>
> To empirically examine this effect, we conduct ***Random Selection*** on the retain set by randomly selecting a subset of non-target concepts to study the performance under different retain-set scales. We evaluate this by erasing *Van Gogh*, using its CS to measure erasure efficacy and the average FID over the four artistic styles (i.e., Picasso, Monet, Paul Gauguin, and Caravaggio) to measure prior preservation. As shown in the table below, decreasing the retain-set scale consistently improves erasure efficacy because the expanded null space provides greater degrees of freedom for removing the target concept. In contrast, the FID first decreases and then increases, which indicates that neither an excessively large nor an excessively small retain set can maintain prior knowledge well. Therefore, adjusting the retain-set size achieves a better trade-off between erasure and preservation (e.g., at 46%) compared with using the full retain set (i.e., 100%). **However, manually tuning the retain-set size for each deployment is impractical in real-world use.**
>
> Instead, our IPF module refines this heuristic process by identifying and retaining only the non-target concepts that are most affected by erasure. As a result, the refined retain set neither collapses the null space nor includes unnecessary concepts. As shown in the table, under the same retain-set scales, IPF consistently achieves both better erasure (lower CS) and better prior preservation (lower FID) than Random Selection, demonstrating the effectiveness and generalization ability of IPF.
>
> | **Retain Set Scale** | 100% | 77% | 46% | 20% | 9% |
> | --- | --- | --- | --- | --- | --- |
> |  | CS↓/ FID↓ | CS↓/ FID↓ | CS↓/ FID↓ | CS↓/ FID↓ | CS↓/ FID↓ |
> | **Random Selection** | 27.20/48.19 | 27.07/45.36 | 26.64/41.27 | 26.15/49.03 | 25.82/62.09 |
> | **IPF (Ours)** | 27.20/48.19 | 26.90/44.38 | 26.29/**29.35** | 25.90/34.61 | **25.73**/37.29 |
> | **Improvement** | - | 0.17/0.98 | 0.35/11.92 | 0.25/14.42 | 0.09/24.80 |

---

### Official Review · Reviewer_cb18 · 2025-10-31

**Soundness:** 3
**Presentation:** 3
**Contribution:** 3
**Rating:** 6
**Confidence:** 3

**Summary:**

This paper proposes SPEED, which resolves the issue of reduced generation quality of non-target concepts due to conflicts in optimization objectives through three modules, Influence-based Prior Filtering (IPF), Directed Prior Augmentation (DPA) and Invariant Equality Constraints (IEC). Extensive experiments have proved that the method is scalable, accurate and efficient.

**Strengths:**

1. The paper is well written with clear presentations, and easy to follow.
2. Extensive experiments show that SPEED consistently outperforms existing methods in prior preservation across various erasure tasks.

**Weaknesses:**

1. The proposed method achieves a lower CS compared to RECE. The authors argue that an excessively low CS, as seen in RECE, results in "over-erasure", whereas the higher CS of SPEED is deemed "adequate." How can it be proved that the residual concept has no negative impact on the erasing effect of the model?
2. Is the IPF sensitive to the initial selection of the retention set? If the retention set is randomly sampled, is the IPF still valid?
3. Please supplement the experiments to illustrate that SPEED has the lowest computational cost and high operational efficiency.

**Questions:**

Please refer to the weaknesses.

---

> ### Author Response · Authors · 2025-11-21
> **Author Response (Part 1/3)**
>
> > W1: The proposed method achieves a lower CS compared to RECE. The authors argue that an excessively low CS, as seen in RECE, results in "over-erasure", whereas the higher CS of SPEED is deemed "adequate." How can it be proved that the residual concept has no negative impact on the erasing effect of the model?
> >
>
> We thank the reviewer for the thoughtful comments.
>
> We acknowledge that the phrasing "lower CS values typically indicate over-erasure" in Line 374 was ambiguous. We did not intend to claim that a low CS inherently means “over-erasure” of the target concept. Instead, we use “over-erasure” to describe the observation that excessively suppressing CS, as seen in RECE, often harms the model’s generative capability for other non-target concepts, leading to degraded prior preservation. We have revised this part in the main text (line 373-375).
>
> As discussed in Appx. D.1 (Fig. 8), in the Snoopy case, our method reduces the similarity with “Snoopy” in the original image from CS = 30.35 to CS = 24.18, which already achieves visually successful erasure (Snoopy $\to$ dog). In contrast, RECE further lowers CS to 19.79 by additionally suppressing dog-related semantics (Snoopy $\to$ landscape), at the cost of a worse FID on non-target concepts (25.61 vs. our 19.95 in Table 1, MS-COCO FID). In Table 2, we show that as the number of target concepts increases, the deterioration of non-target concepts gradually accumulates. RECE’s MS-COCO CS drops from 16.75 (erasing 10 concepts) to 13.49 (erasing 50 concepts) and further to 12.09 (erasing 100 concepts), which generates nearly meaningless noisy images for general concepts. In contrast, our method achieves MS-COCO CS from 26.47 (erasing 10 concepts) to 26.46 (erasing 50 concepts) to 26.22 (erasing 100 concepts), achieving a superior balance between erasure efficacy and prior preservation.
>
> We illustrate that our erasure is already sufficient in Figs. 4-7. To further demonstrate that our current erasure is “adequate”, we additionally conduct a human study. For our method and RECE, we randomly sample 50 generated images per method to erase *Snoopy and Mickey, and Spongebob,* and ask human evaluators to determine whether the target concept’s semantics remain. We recruited 30 human participants through Amazon Mechanical Turk to vote “yes” or “no” on whether the concept is visually erased or not. The final erasure success rates (%) are reported in the table below. The overall results (RECE’s 98.76% v.s. Our 98.47%) indicate that our method achieves successful erasure on par with RECE from the human perspective. The human study details and final results are added to our revised paper in Appx. D.1
>
> | **Erasure Success Rate (%)** | ***Erase ~~Snoopy~~ and ~~Mickey~~ and ~~Spongebob~~*** |  |  |  |
> | --- | --- | --- | --- | --- |
> |  | Snoopy | Mickey | Spongebob | Average |
> | **RECE** | 98.93% | 98.27% | 99.07% | 98.76% |
> | **Ours** | 98.20% | 98.40% | 98.80% | 98.47% |
>
> Considering strong erasure efficacy is necessary in certain scenarios (e.g., adversarial attacks), in Appx.D.4, we further introduce a variant of our method (i.e., SPEED w/ AT), which incorporates the adversarial training (AT) strategy from RECE into our method. As shown in Table 8, SPEED w/ AT achieves better performance than RECE in all benchmarks, and achieves comparable robustness to state-of-the-art methods such as CPE, AdvUnlearn, RACE, and Receler.

---

> ### Author Response · Authors · 2025-11-21
> **Author Response (Part 2/3)**
>
> > W2: Is the IPF sensitive to the initial selection of the retention set? If the retention set is randomly sampled, is the IPF still valid?
> >
>
> Thank you for this insightful question. **Our IPF module consistently improves erasure performance regardless of the retain set choice.** This robustness stems from the nature of null-space constrained editing. An overly large retain set increases the rank of the associated feature matrix, which shrinks the null space and undermines prior preservation. IPF alleviates this issue by identifying and retaining only the most affected non-target concepts, whose representations are most susceptible to distortion. This selective refinement stabilizes the null space and enhances preservation across different initializations of the retain set.
>
> To empirically validate this, we perform additional experiments on artistic style erasure (erasing *Van Gogh*) using different retain sets: a randomly sampled retain set from MS-COCO of different scales and our default retain set of 1,734 artistic styles following UCE. The results show that **IPF is effective in all cases**, consistently improving erasure–preservation trade-off by identifying the most affected non-target concepts and discarding weakly relevant ones. Moreover, **the improvement is more pronounced with the targeted retain set**, because erasing an artistic style induces larger prior shifts on semantically similar styles, enabling IPF to more accurately capture the concepts that require preservation. These results confirm that IPF provides a robust and adaptive refinement of any given retain set.
>
> | **Erase *~~Van Gogh~~*** |  | **Van Gogh** | **Picasso** | **Monet** | **Paul Gauguin** | **Caravaggio** |
> | --- | --- | --- | --- | --- | --- | --- |
> |  |  | CS ↓ | FID ↓ | FID ↓ | FID ↓ | FID ↓ |
> | **Random (1K)** | w/o IPF | 26.06 | 73.78 | 82.18 | 84.73 | 89.52 |
> |  | w/ IPF | 25.94 (-0.12) | 71.66 (-2.12) | 79.70 (-2.48) | 75.34 (-9.39) | 88.51 (-1.01) |
> | **Random (2K)** | w/o IPF | 26.11 | 89.12 | 82.81 | 81.91 | 92.08 |
> |  | w/ IPF | **25.93 (-0.18)** | 75.26 (-13.86) | 80.63 (-2.18) | 76.93 (4.98) | 88.47 (-3.61) |
> | **Random (3K)** | w/o IPF | 26.36 | 83.67 | 85.92 | 89.79 | 89.68 |
> |  | w/ IPF | 25.95 (-0.41) | 77.80 (-5.87) | 83.31 (-2.61) | 81.93 (-7.86) | 89.53 (-0.15) |
> | **Targeted (1.7K)** | w/o IPF | 26.79 | 45.36 | 30.06 | 31.89 | 54.92 |
> |  | w/ IPF | 26.29 (-0.50) | **35.86 (-9.50)** | **16.85 (-13.21)** | **24.94 (-6.95)** | **39.74 (-15.17)** |

---

> ### Author Response · Authors · 2025-11-21
> **Author Response (Part 3/3)**
>
> > W3: Please supplement the experiments to illustrate that SPEED has the lowest computational cost and high operational efficiency.
> >
>
> Thank you for your comment. The computational cost and operational efficiency of SPEED have already been compared in Table 2. Here we further provide a more detailed comparison, highlighting how SPEED consistently achieves low computational overhead while maintaining high erasure performance across all evaluated settings.
>
> As shown in the Table below, **SPEED achieves consistently low runtime across all multi concept erasure settings while maintaining strong erasure efficacy and prior preservation across all multi-concept erasure settings.** For instance, SPEED erases 100 target concepts in 5 seconds with $H_o=89.6$ on a single A100 GPU, whereas the state-of-the-art multi-concept erasure baseline MACE requires about 30 minutes with $H_o=87.1$ under the same hardware setup, and training-based methods such as ConAbl require around 150 minutes with $H_o=58.0$ only. UCE and RECE are indeed faster than training-based approaches and also operate in a few seconds. However, their performance deteriorates significantly as the number of concepts increases. For example, when erasing 100 concepts, UCE requires only 2.1 seconds but its $H_o$ drops from 83.1% (erasing 10 concepts) to 34.6% (erasing 100 concepts). RECE requires 11 seconds, but its $H_o$ drops from 80.4% (erasing 10 concepts) to 38.2% (erasing 100 concepts). Moreover, both UCE and RECE achieve MSCOCO CS at 185.46 and 177.57 (from Table 2 in our main paper), indicating substantial degradation of general knowledge and an inability to reliably achieve erasure-preservation balance under multi-concept erasure scenarios, which is more practical in real-world applications.
>
> |  |  | **Training-based** |  | **Editing-based** |  |  |
> | --- | --- | --- | --- | --- | --- | --- |
> |  |  | ConAbl | MACE | UCE | RECE | Ours |
> |  | **Data Preparation Time** | $n \times 1000 t_1$ | $n \times (8 t_1 + 8 t_2)$ | 0 | 0 | 0 |
> | **~~10 concept~~** | **Erasure Time** | $10 \times 90$ | 207.0 | 1.5 | 2.5 | 3.8 |
> |  | **Performance** $\boldsymbol{H_o}$ ↑ | 52.2 | 92.7 | 83.1 | 80.4 | **93.4** |
> | **~~50 concepts~~** | **Erasure Time** | $50 \times 90$ | 936.0 | 1.8 | 6.3 | 4.2 |
> |  | **Performance** $\boldsymbol{H_o}$ ↑ | 48.7 | 90.0 | 48.4 | 33.0 | **92.3** |
> | **~~100 concepts~~** | **Erasure Time** | $100 \times 90$ | 1735.9 | 2.1 | 11.0 | 5.0 |
> |  | **Performance** $\boldsymbol{H_o}$ ↑ | 58.0 | 87.1 | 34.6 | 38.2 | **89.6** |
>
> **Duration comparison (s) in erasing multiple celebrities** on one A100 GPU, where $n$ is the number of target concepts. During data preparation, ConAbl requires pre-sampling 1,000 images ($t_1$ per image) for each target concept while MACE needs 8 pre-sampled images along with 8 corresponding segmentation masks ($t_2$ per mask) using SAM.

---

### Official Review · Reviewer_gLg4 · 2025-11-01

**Soundness:** 3
**Presentation:** 3
**Contribution:** 3
**Rating:** 8
**Confidence:** 4

**Summary:**

This paper proposes an efficient and effective concept erasure method, SPEED. Inspired from the null-space constraints, SPEED edits model weights within the null space of non-target concepts.
However, as the number of non-target concepts grows, finding the appropriate null space becomes challenging. To address this, the paper introduces Influence-based Prior Filtering (IPF), which removes the minimally affected non-target concepts from the retain set. Then, to further enhance preservation, it introduces Directed Prior Augmentation (DPA), which augments the remaining non-target concepts with semantically consistent noise. Finally, the paper proposes Invariant Equality Constraints (IEC) to ensure that the [SOT] token and the null-text embedding remain unchanged during unlearning.

**Strengths:**

- The paper provides a theoretical analysis showing that the closed-form optimization used in UCE is not optimal, as its preservation error has a non-zero bound, which limits non-target preservation as the number of target concepts increases.
- The proposed method is effective in non-target preservation and achieves a good balance between concept erasure and non-target preservation for multi-concept erasure.
- The proposed method is efficient, which only needs 5 seconds to erase 100 concepts on a single A100 GPU.
- The experiments are comprehensive, covering a wide range of domains including copyrighted content, artistic styles, celebrities, and nudity.
- The ablation studies are detailed. The paper investigates which matrices in cross-attention layers should be modified, the effect of the filtering threshold in IPF, and the effect of augmentation times $N_A$ and ranks $r$ in DPA.

**Weaknesses:**

- The proposed method is less effective in erasing target concepts. In the copyrighted content erasure results (Table 1 left), SPEED’s CLIP scores are much higher than those of prior methods RECE and MACE. Similarly, in the multi-concept erasure experiments (Table 2), SPEED’s erasure accuracies are also higher than RECE and MACE, indicating weaker erasure performance compared to these baselines.

**Questions:**

None

---

> ### Author Response · Authors · 2025-11-21
> **Author Response**
>
> > W1: The proposed method is less effective in erasing target concepts. In the copyrighted content erasure results (Table 1 left), SPEED’s CLIP scores are much higher than those of prior methods RECE and MACE. Similarly, in the multi-concept erasure experiments (Table 2), SPEED’s erasure accuracies are also higher than RECE and MACE, indicating weaker erasure performance compared to these baselines.
> >
>
> We thank the reviewer for the thoughtful comments. Our response focuses on the following three key points:
>
> (1) While our method does not always achieve the lowest CLIP Score (CS) on target concepts in instance erasure, we illustrate that our erasure is already sufficient in Figs. 4-7. As further discussed in Appx. D.1 (Fig. 8), RECE reaches lower CS by suppressing the other semantics (e.g., removing *Snoopy* into a landscape without a subject, and changing *Mickey* into a generic person), but this comes at the cost of noticeable degradation in prior preservation (by comparing RECE and UCE in Table 1). To further demonstrate that our current erasure is adequate, we additionally conduct a human study. For our method and RECE, we randomly sample 50 generated images per method to erase *Snoopy and Mickey, and SpongeBob,* and ask human evaluators to determine whether the target concept’s semantics remain. We recruited 30 human participants through Amazon Mechanical Turk to vote “yes” or “no” on whether the concept is visually erased or not. The final erasure success rates (%) are reported in the table below. The overall results (RECE’s 98.76% v.s. Our 98.47%) indicate that our method achieves successful erasure on par with RECE from the human perspective. The human study details and final results are added to our revised paper in Appx. D.1.
>
> | **Erasure Success Rate (%)** | ***Erase ~~Snoopy~~ and ~~Mickey~~ and ~~Spongebob~~*** |  |  |  |
> | --- | --- | --- | --- | --- |
> |  | Snoopy | Mickey | Spongebob | Average |
> | **RECE** | 98.93% | 98.27% | 99.07% | 98.76% |
> | **Ours** | 98.20% | 98.40% | 98.80% | 98.47% |
>
> (2) Our method, SPEED, is designed to be **scalable**, **precise**, and **efficient**, which provides clear advantages over existing approaches.
>
> - Compared with RECE, our method is superior primarily in scalability and precision. RECE’s adversarial editing paradigm indeed achieves effective erasure, but this comes at the cost of degrading prior preservation. As the number of target concepts increases, the deterioration of prior preservation gradually accumulates. As shown in Table 2, RECE’s MS-COCO CS drops from 16.75 (erasing 10 concepts) to 13.49 (erasing 50 concepts) and further to 12.09 (erasing 100 concepts), which generates nearly meaningless noisy images for general concepts. In contrast, our method achieves MS-COCO CS from 26.47 (erasing 10 concepts) to 26.46 (erasing 50 concepts) to 26.22 (erasing 100 concepts), while still ensuring strong prior preservation after erasing 100 concepts.
> - Compared with MACE, the current SOTA multi-concept erasure method, our advantage lies in both efficiency and multi-concept performance. MACE requires about **30 minutes** to erase 100 concepts with a harmonic mean score of $H_o = 87.1$, while our closed-form solution enables direct computation of the model update and achieves erasure of 100 concepts in **only 5 seconds** with $H_o = 89.6$.
>
> (3) Considering strong erasure efficacy is necessary in certain scenarios (e.g., adversarial attacks), in Appx. D.4, we further introduce a variant of our method (i.e., SPEED w/ AT), which incorporates the adversarial training (AT) strategy from RECE into our method. As shown in Table 8, without adversarial training, SPEED w/o AT already outperforms other non–adversarial-training baselines (e.g., UCE); after applying adversarial training, SPEED w/ AT surpasses RECE and MACE in all benchmarks, and achieves comparable robustness to state-of-the-art methods such as CPE, AdvUnlearn, RACE, and Receler, maintaining substantially lower runtime at only 4.5 seconds.

---

### Author Response · Authors · 2025-11-23
**Summary of Paper Revision**

We gratefully thank all the reviewers for their positive ratings and constructive feedback. We are encouraged that they find our proposed method, **SPEED**, to be scalable, precise, and efficient (`NrN5`, `gLg4`, `oMhk`), and technically sound with valuable theoretical analysis regarding the limitations of previous closed-form solutions (`gLg4`). We are also glad that they acknowledge our extensive experiments demonstrate superior performance in non-target preservation (`gLg4`, `cb18`, `oMhk`) with detailed ablations (`gLg4`) and that the paper is well-organized with clear presentations (`cb18`, `oMhk`).

We address the concerns and questions in detail below. According to these comments, we have improved our manuscript (modifications are marked in **red**) and summarized the main changes as follows:

1. **Human evaluation on erasure efficacy (Appx. D.1, to `gLg4`, `cb18`, & `oMhk`):** To address concerns regarding the correlation between CLIP Scores (CS) and erasure success, we conducted a human study with 30 participants. The results demonstrate that SPEED achieves successful erasure rates (98.47%) comparable to state-of-the-art methods like RECE (98.76%), confirming that our method effectively removes target concepts.
2. **Clarification on "over-erasure" (Sec. 5.1, to `cb18`):** We revised the main text (Lines 373-375) to clarify the definition of "over-erasure." We specify that this term describes the observation where excessive suppression of CS harms the model's generative capability for non-target concepts (e.g., causing MS-COCO CS to drop significantly as seen in RECE). We emphasize that SPEED strikes a superior balance by achieving visually successful erasure without such degradation.
3. **Additional ablation studies on retain sets (Appx. D.5, to `gLg4` & `oMhk`):** We included supplementary experiments comparing our Influence-based Prior Filtering (IPF) against random selection and analyzing the impact of retention set scales. These results confirm the robustness of IPF in stabilizing the null space and improving the erasure-preservation trade-off regardless of the initial retention set.
4. **Extended visualizations and applications (Appx. E.1 & E.2, to `NrN5`):** We added detailed qualitative results (Fig. 11) to demonstrate that SPEED preserves non-target semantics (e.g., backgrounds) precisely as specified in the prompts. Furthermore, we extended our method to multi-concept knowledge editing (Fig. 12), showcasing its versatility beyond erasure.

---

### Author Response · Authors · 2025-12-03
**General Response and Summary of Discussion**

We sincerely thank the Area Chair and all reviewers for their time and constructive feedback. We are encouraged that **all reviewers (`gLg4`, `cb18`, `oMhk`, `NrN5`) gave positive ratings (8666, Avg: 6.5)**, recognizing our proposed SPEED method as **scalable, precise, and efficient** in concept erasure. We are particularly grateful that Reviewer `gLg4` championed our work with **an accept rating of 8 (accept, good paper),** highlighting our theoretical soundness and efficiency.

During the discussion period, we actively engaged with all reviewers to address concerns regarding erasure efficacy, method robustness, and module analysis. We further provided extensive new experiments, including a human evaluation study, adversarial robustness, and runtime comparisons. Below is a summary of the discussion.

| **Reviewer** | **Rating** | **Strengths** | **Questions** | **Our Reply** | **Feedback** |
| --- | --- | --- | --- | --- | --- |
| **`gLg4`** | **8: Accept** | **(1)** **Theoretically sound** analysis. **(2)** **Good balance** of erasure and preservation. **(3)** **Highly efficient** (5s for 100 concepts). **(4)** **Comprehensive** experiments. **(5)** **Detailed** ablation studies. | **Erasure Efficacy:** CLIP scores (CS) are higher than baselines (RECE). | (1) We clarified that our erasure is already sufficient through massive qualitative experiments and conducted a Human Study showing our success rate (**98.47%**) is on par with RECE (**98.76%**). (2) We demonstrated our distinct advantages in being **scalable, precise, and efficient** over SOTA methods (RECE and MACE). (3) We introduced a variant, SPEED w/ AT, to demonstrate that our method can also achieve strong erasure efficacy if required. | **No response** |
| **`cb18`** | **6: BA** | **(1)** **Consistently outperforms** in prior preservation. **(2)** **Clear presentation.** **(3)** **Easy to follow.** | **(1) Justification:** Does the residual concept have negative impact on erasure? **(2) Discussion:** Is IPF robust to initial retention set choice? **(3) Efficiency:** Need detailed cost comparisons. | **(1)** We clarified the definition of “over-erasure” and, through case analysis and a human study, showed that the residual concept has no negative impact while our method still achieves adequate erasure compared to RECE. **(2)** We provided new ablations with different retain sets, showing IPF's consistent improvement in all cases. **(3)** We added a detailed comparison confirming SPEED is significantly faster (5s) than SOTA methods (MACE: ~30 mins) in erasure without any data preparation time. | **No response** |
| **`oMhk`** | **6: BA** | **(1)** **Strong ability** to preserve untargeted concepts. **(2)** **High computational efficiency** via closed-form solution. **(3)** The paper is **well-organized.** | **(1) Erasure Efficacy:** CLIP scores (CS) are higher than baselines (RECE). **(2) Discussion:** Can adjusting the number of non-target concepts (i.e., retain-set size) improve the erasure-preservation trade-off? | **(1)** Same as the response to `gLg4`.  **(2)** We confirmed that manually adjusting the retain-set size can improve the trade-off, while our IPF module achieves consistently better erasure and preservation than such manual adjustment under the same retain-set size. | **No response** |
| **`NrN5`** | **6: BA** | **(1)** Focus on **multiple crucial aspects:** scalability, precision, and efficiency. **(2)** Method framework is **reasonable and technically sound.** **(3)** Easy to follow and **well-organized.** | **(1) Robustness:** Can SPEED handle paraphrased prompts? **(2) Background Change:** Does erasure alter background content? **(3) More application:** Can it do multi-concept editing? | **(1)** We added Robustness Evaluations (Appx D.4) showing SPEED (w/ Adversarial Training) resists paraphrased prompts effectively. **(2)** We clarified that background changes occur because no background was specified, and provided visuals (Fig. 11) proving that our erasure has no impact on other non-target semantics in the prompt. **(3)** We added Fig. 12 demonstrating successful multi-concept knowledge editing. | **No response** |

---

### Meta-Review · Area_Chair_Lu5u · 2026-01-04

**Summary:**

This paper proposes SPEED, a parameter-editing-based approach for scalable and precise concept erasure in diffusion models. While initial reviews raised concerns about comparatively weaker erasure strength as measured by CLIP-based metrics, the authors have provided a thorough and convincing revision. In particular, extensive qualitative results, a large-scale human study demonstrating parity with SOTAs (e.g., RECE), and a SPEED w/ AT variant confirm that the framework can achieve strong erasure when required. The authors also justify their notion of “adequate erasure” versus over-erasure through case analyses and user studies, showing no negative impact from residual signals. Additional ablations validate the robustness of IPF across retain-set choices, while new robustness experiments demonstrate resistance to paraphrased prompts. In addition, runtime comparisons substantiate SPEED’s significant efficiency advantage over prior methods, reinforcing its scalability claims. Overall, the AC believes the rebuttal responses satisfactorily address all major concerns, and the paper makes a strong and well-supported contribution to practical concept erasure in large diffusion models.

**Reviewer Concerns:**

1. Insufficient Target Concept Erasure
2. Justification of “Adequate” vs. “Over-Erasure” (Evaluation Ambiguity)
3. Robustness to Paraphrased Prompts (Concept Recoverability)
4. Impact on Non-Target Visual Regions (Side Effects)
5. Sensitivity and Validity of IPF (Methodological Robustness)
6. Computational Efficiency Claims Need Empirical Support

**Reviewer Scores:**

1. The added human study closes the gap between quantitative CLIP-based criticism and perceived erasure effectiveness.
2. The SPEED w/ AT variant convincingly rebuts the claim that the framework is intrinsically weak at erasure.
3. The robustness and paraphrase evaluations address recoverability concerns.
4. The IPF ablations resolve methodological sensitivity questions.
5. The explicit runtime comparisons strongly validate the paper’s scalability and efficiency claims.
6. The clarification and empirical grounding of “over-erasure” reframes the evaluation debate from metric minimization to practical usability, which was the central conceptual dispute raised by multiple reviewers.

Given the above rebuttal responses, I believe the reviewers would remain positive about this paper.

---

### Decision · Program_Chairs · 2026-01-26

Accept (Poster)